# Robust Wasserstein $k$-center Clustering: Algorithms and Acceleration

## Abstract

The classical metric $k$-center problem is widely used in data representation tasks. However, real-world datasets often contain noise and exhibit complex structures, making the traditional metric $k$-center problem insufficient for such scenarios. To address these challenges, we present the **R**obust **W**asserstein **C**enter clustering (RWC-clustering) problem. Compared to the classical setting, the main challenge in designing an algorithm for the RWC-clustering problem lies in effectively handling noise in the cluster centers. To this end, we introduce a dedicated purification step to eliminate noise, based on which we develop our customized clustering algorithms. Furthermore, when dealing with large-scale datasets, both storage and computation become highly resource-intensive. To alleviate this, we adopt the *coreset* technique to improve the computational and storage efficiency by compressing the dataset. Roughly speaking, this coreset method enables us to compute the objective value on a small-size coreset, while ensuring a close approximation to the value on the original dataset in theory; thus, it substantially saves the storage and computation resources. Finally, experimental results demonstrate the effectiveness of our RWC-clustering problem and the efficiency of the coreset method.

## 1 Introduction

The metric $k$-center problem [Hakimi, 1964] is widely used in data compression [Łącki et al., 2024] and representation learning [Bateni et al., 2023]. Its objective is to select $k$ centers, forming a $k$-center set $C$, such that the maximum distance from any data point to its closest center is minimized. More formally, for a given dataset $Q$ in metric space $(\mathcal{X}, \text{dist})$, the metric $k$-center problem can be formulated as

$$\min_{C \subseteq \mathcal{X}, |C|=k} \max_{\mu \in Q} \min_{\nu \in C} \text{dist}(\mu, \nu). \qquad \text{(metric } k\text{-center problem)}$$

Data in combinatorial optimization [Luo et al., 2023, Grinsztajn et al., 2023, Drakulic et al., 2023] and biomedical fields [Thual et al., 2022, Bazeille et al., 2019] often exhibit complex structures and are typically represented as probability distributions. Nevertheless, the traditional Euclidean distance falls short in describing the geometric structure of such data. In contrast, the Wasserstein distance [Peyré et al., 2017] excels at capturing the geometric structure, making it a powerful tool for quantifying the difference between these complex data items.

However, the real-world datasets are often contaminated by noise, and the Wasserstein distance is sensitive to outliers [Nietert et al., 2022] due to its stringent marginal constraints. Specifically, even a single outlier with negligible mass can substantially distort the final result by adjusting its position, thereby limiting its utility in practical scenarios. To address this issue, we adopt the Robust Wasserstein Distance (RWD) [Nietert et al., 2022] to measure the similarity between the data items. Based on this, we introduce the **R**obust **W**asserstein **C**enter clustering (RWC-clustering) problem (in Definition 2.1) to effectively represent these complex datasets.

Submitted to 39th Conference on Neural Information Processing Systems (NeurIPS 2025). Do not distribute.

Solving the RWC-clustering is a typical non-convex optimization problem. Initialization is crucial for non-convex optimization, as it directly affects whether the optimization algorithm can escape the local minima and effectively find the global optimum. In the classical metric $k$-center problem, Gonzalez's algorithm [Gonzalez, 1985] is often used as a seeding algorithm to provide a good initialization; a local search algorithm [Lattanzi and Sohler, 2019, Choo et al., 2020] is then used as a post-processing step to further refine the solution.

In the classical setting, both algorithms [Gonzalez, 1985, Choo et al., 2020] select data points directly from the original dataset as cluster centers. However, in our noisy setting, selecting candidates from the noise-contaminated dataset inevitably leads to noisy candidate centers. This contradicts our goal of obtaining clean cluster centers. To address this issue, we introduce a dedicated purification step to remove noise from the candidate centers, thereby producing clean centers. We then plug this purification step into existing algorithms, designing tailored initialization and post-processing procedures for our RWC-clustering problem.

Except for algorithm design, scalability is also a key consideration. When handling large datasets, solving the RWC-clustering problem becomes extremely time-consuming and requires significant storage resources. To address this issue, we introduce *coreset* [Ros and Guillaume, 2020], a widely used data compression technique. A coreset can be regarded as a summary of the original dataset with respect to certain objective; it enhances computational and storage efficiency by reducing the dataset size. Roughly speaking, it enables us to approximate the value computed on the original dataset by the value on a small-size coreset. Thus, it helps save computational and storage resources substantially while maintaining accuracy closely.

Although many coreset techniques[Huang et al., 2024, Huang et al., 2023] have been developed for the classical clustering problems, they are primarily designed for metric spaces. However, RWD is not a metric; thus, although existing techniques may provide useful insights, new theoretical analysis is still necessary for the design of our coreset.

**Our contributions:**

- For effectively representing datasets with complex structures and outliers, we introduce the RWC-clustering problem and provide the underlying intuition for its formulation.

- To solve this robust clustering problem, we first design a purification step to eliminate the noise in the candidate centers; then, we integrate it into existing methods [Gonzalez, 1985, Lattanzi and Sohler, 2019, Choo et al., 2020] to develop customized initialization and post-processing algorithms for our RWC-clustering problem.

- Furthermore, to enhance scalability, we introduce the coreset technique to accelerate the computation by compressing the dataset. Additionally, we theoretically demonstrate that the coreset is a good proxy of the original dataset.

- Finally, we experimentally demonstrated the effectiveness of our RWC-clustering problem and the efficiency of the coreset method.

## 1.1 Other related works

**Optimal transport (OT)** is a popular tool for quantifying the difference between probability measures. Several algorithms have been developed for solving the OT problem. Peyré et al. [2017] introduced an $\epsilon_+$-approximation algorithm by using the interior point method within $\widetilde{\mathcal{O}}(n^3)$ time, where $\epsilon_+$ denotes the additive error and $n$ is the support size of measures. Subsequently, Dvurechensky et al. [2018] proposed the Sinkhorn's algorithm, which reduces the time complexity to $\widetilde{\mathcal{O}}(n^2/\epsilon_+^2)$ by solving the entropic regularization version [Cuturi, 2013]. Especially, Jambulapati et al. [2019] further improved this result by leveraging the area-convexity and dual extrapolation techniques, achieving $\widetilde{\mathcal{O}}(n^2/\epsilon_+)$ time complexity.

**Gonzalez's algorithm** [Gonzalez, 1985], a 2-approximation algorithm for the metric $k$-center problem, is often used as an initialization method in clustering tasks. It iteratively selects the point farthest from the currently chosen centers as the new center. The sequential nature of center selection leads to dependencies between steps, which poses challenges for achieving parallel computation. It takes $\mathcal{O}(mk)$ time, where $m$ is the size of the dataset, and $k$ represents the number of centers. When $k$ or $m$ is large, the computational complexity becomes a bottleneck.

**Hierarchical Gonzalez's algorithm** [Murtagh and Contreras, 2012] is a variation of the Gonzalez's algorithm tailored to address hierarchical clustering problems. This algorithm constructs a tree structure by recursively splitting data at different levels of granularity. It selects cluster centers sequentially within localized regions using the Gonzalez's algorithm while incorporating a globally parallelizable design, resulting in high efficiency.

## 2 Preliminaries

**Notations:** We adopt some notation conventions from [Nietert et al., 2022, Wang et al., 2024]. We define $[n] := \{1, \ldots, n\}$. Let $(\mathcal{X}, \text{dist})$ be a metric space and $\mathbb{R}_+$ be the set of non-negative real numbers. We use $\mathcal{M}_+(\mathcal{X})$ to denote the positive measure space on $\mathcal{X}$, and $\mathcal{P}(\mathcal{X})$ the corresponding probability measure space.

Matrices are denoted by capital boldface letters, such as $\mathbf{P}$; $P_{ij}$ denotes its element in the $i$-th row and $j$-th column. Similarly, vectors are represented by lowercase boldface letters, such as $\mathbf{a} := (a_1, \ldots, a_d)^T \in \mathbb{R}^d$; $a_i$ is its $i$-th element. Let $|Q|$ be the cardinality of the set $Q$. For measures $\mu', \mu \in \mathcal{M}_+(\mathcal{X})$, the notation $\mu' \leq \mu$ means that $\mu'(A) \leq \mu(A)$ for any set $A \subseteq \mathcal{X}$.

**Wasserstein distance:** Let $\mu = \sum_{i=1}^n a_i \delta_{x_i}, \nu = \sum_{j=1}^n b_j \delta_{y_j}$ be two discrete probability measures[1] in $\mathcal{P}(\mathcal{X})$, where $\mathbf{a} = (a_1, \ldots, a_n)^T, \mathbf{b} = (b_1, \ldots, b_n)^T \in \mathbb{R}_+^n$ are their weight vectors and $\delta$ is the Dirac delta function. Given any real number $z \geq 1$ and a cost matrix $\mathbf{D} \in \mathbb{R}_+^{n \times n}$ with $D_{ij} = \text{dist}^z(x_i, y_j)$, the $z^{\text{th}}$-Wasserstein distance between $\mu$ and $\nu$ is defined as

$$W(\mu, \nu) := \left( \min_{\mathbf{P} \in \Pi(\mathbf{a}, \mathbf{b})} \langle \mathbf{P}, \mathbf{D} \rangle \right)^{1/z}, \tag{1}$$

where $\Pi(\mathbf{a}, \mathbf{b}) := \left\{ \mathbf{P} \in \mathbb{R}_+^{n \times n} \mid \mathbf{P}\mathbf{1} = \mathbf{a}, \mathbf{P}^T\mathbf{1} = \mathbf{b} \right\}$ is the set of all feasible couplings, $\mathbf{1}$ is the vector of all ones, and $\langle \mathbf{P}, \mathbf{D} \rangle$ denotes the Frobenius inner product between $\mathbf{P}$ and $\mathbf{D}$.

Optimal Transport (OT) shares a similar formulation with Wasserstein distance, but their cost matrices differ. The cost matrix in OT is derived from a positive function. In contrast, the cost matrix for Wasserstein distance has stricter requirements—it must be induced by a distance function. Thus, the Wasserstein distance is a metric, while OT is not necessarily one. Despite these differences, OT algorithms can still be effectively used to compute Wasserstein distance.

**Robust Wasserstein distance:** Although the Wasserstein distance [Villani et al., 2009, Peyré et al., 2017] is widely used for measuring the difference between two probability measures, its sensitivity to outliers limits its applicability in noisy scenarios. To overcome this limitation, several robust variants [Nietert et al., 2022, Le et al., 2021, Chapel et al., 2020] have been proposed. This paper focuses on the following robust version.

**Definition 2.1** (Robust Wasserstein distance [Nietert et al., 2022, Wang et al., 2024])**.** *Let $\mu$ and $\nu$ be the same as in Equation* (1)*. Given two pre-specified parameters $0 \leq \zeta_\mu, \zeta_\nu < 1$, the robust Wasserstein distance $\widetilde{\mathcal{W}}(\mu, \nu)$ between $\mu$ and $\nu$ is formulated as*

$$\widetilde{\mathcal{W}}(\mu, \nu) := \min_{\substack{\mu', \nu' \in \mathcal{M}_+(\mathcal{X}) \\ \mu' \leq \mu, \|\mu - \mu'\|_{TV} = \zeta_\mu \\ \nu' \leq \nu, \|\nu - \nu'\|_{TV} = \zeta_\nu}} W\left( \frac{\mu'}{1 - \zeta_\mu}, \frac{\nu'}{1 - \zeta_\nu} \right), \tag{2}$$

*where $\| \cdot \|_{TV}$ denotes the total variation (TV) norm.*

Moreover, Equation (2) can be reformulated as an (augmented) OT problem [Wang et al., 2024] by introducing a dummy point, allowing it to be solved efficiently by using the existing OT solvers.

**Note:** Henceforth, we denote the Wasserstein distance between $\mu$ and $\nu$ by $W(\mu, \nu)$. The notation $\widetilde{\mathcal{W}}(\mu, \nu)$ represents the robust Wasserstein distance when both $\mu$ and $\nu$ contain $\zeta$ mass of outliers.

Specially, $\mathcal{W}(\mu, \nu)$ refers to the case where $\mu$ contains $\zeta$ mass of outliers while $\nu$ has no outliers.

---

[1]To simplify the expression, the support size of all measures in this paper is set to $n$.

**Robust clustering:** We propose a robust version of the Wasserstein $k$-center clustering problem. Its goal is to cover all data points using $k$ balls of equal radius under robust Wasserstein distance $\mathcal{W}(\cdot, \cdot)$, while minimizing the radius of these balls.

**Definition 2.2** (RWC-clustering). *Given a set of probability measures $Q = \left\{\mu^i\right\}_{i \in [m]} \subseteq \mathcal{P}(\mathcal{X})$, the $k$-RWC-clustering problem is to find a $k$-center set $C \subseteq \mathcal{P}(\mathcal{X})$ with $|C| = k$ such that the following objective is minimized.*

$$\mathsf{cost}(Q, C) := \max_{\mu \in Q} \mathcal{W}(\mu, C),$$

*where $\mathcal{W}(\mu, C) := \min_{\nu \in C} \mathcal{W}(\mu, \nu)$.*

The input data points in $Q$ contain outliers. However, our goal is to obtain clean cluster centers. Thus, we define the RWC-clustering problem using $\mathcal{W}(\cdot, \cdot)$ instead of $\widetilde{\mathcal{W}}(\cdot, \cdot)$. Further illustrations are provided in Section 3.

**Coreset:** When the dataset is large, both computation and storage become resource-intensive. To address this issue, we introduce, coreset, a popular data compression technique.

**Definition 2.3** (Coreset). *Given a set of probability measures $Q = \left\{\mu^i\right\}_{i \in [m]} \subseteq \mathcal{P}(\mathcal{X})$ and a real number $\epsilon > 0$, a set $S$ is an $\epsilon$-coreset for the $k$-RWC-clustering problem on $Q$, if the following inequality holds for all $k$-center set $C \subseteq \mathcal{P}(\mathcal{X})$.*

$$|\mathsf{cost}(Q, C) - \mathsf{cost}(S, C)| \leq \epsilon \cdot \mathsf{cost}(Q, C)$$

Essentially, a coreset is a small proxy of the original dataset. To approximate the objective value, we can execute algorithms on this small-size coreset instead of the full dataset. Overall, this approach significantly reduces computational and storage requirements while preserving the objective value.

**Organization:** This paper is organized as follows. In Section 3, we explain the underlying intuition behind the RWC-clustering problem and design an algorithm to solve it. In Section 4, we introduce the coreset technique to accelerate the computation. Finally, in Section 5, we validate the effectiveness of the proposed methods through experimental results.

# 3 Our intuition and algorithms

This section provides the intuition behind using $\mathcal{W}(\cdot, \cdot)$ to measure the difference between data points and cluster centers in RWC-clustering problem. We also introduce a tailored initialization algorithm (see Algorithm 1) and a post-processing algorithm (see Algorithm 4 in the appendix) for this robust setting.

**Intuition of using $\mathcal{W}(\cdot, \cdot)$ in RWC-clustering problem:** In our RWC-clustering problem, we essentially replace the metric $\mathrm{dist}(\cdot, \cdot)$ in metric $k$-center problem with $\mathcal{W}(\cdot, \cdot)$. To illustrate why $\mathcal{W}(\mu, \nu)$ is chosen to measure the distance between a data point $\mu$ and its center $\nu$, rather than using $\widetilde{\mathcal{W}}(\mu, \nu)$, we consider the following example.

**Example 3.1** (Intuition). *Let $x_0 = (0, 0)$ and $x_1 = (0, 1000)$ be two points in $\mathbb{R}^2$. Let $\mu^0 = \delta_{x_0}$ and $\mu^1 = \delta_{x_1}$ be two data points, and let $\nu = 0.5 \cdot \delta_{x_0} + 0.5 \cdot \delta_{x_1}$ be a center. Here, we set $\zeta = 0.5$.*

***Case1:*** *When employing $\widetilde{\mathcal{W}}(\cdot, \cdot)$ to measure the differences, we have $\widetilde{\mathcal{W}}(\mu^0, \nu) = 0$, $\widetilde{\mathcal{W}}(\mu^1, \nu) = 0$ and $\widetilde{\mathcal{W}}(\mu^0, \mu^1) = 1000$. In this case, both $\mu^0$ and $\mu^1$ are contained within a ball of arbitrarily small radius centered at $\nu$ under $\widetilde{\mathcal{W}}(\cdot, \cdot)$. However, the difference between $\mu$ and $\nu$ can be large. In other words, two points within a small ball could exhibit significant differences. Nevertheless, the goal of clustering is to group similar points together. This situation is obviously unreasonable and contradicts the goal of clustering.*

***Case2:*** *In contrast, when using $\mathcal{W}(\cdot, \cdot)$, if both $\mu^0$ and $\mu^1$ lie within a small-radius ball centered at $\nu$, their difference under $\mathcal{W}(\cdot, \cdot)$ remains small. This implies that points within the same small ball exhibit high similarity, which is consistent with the goal of the traditional clustering. (The detailed proofs supporting this claim are provided in Lemma C.1.)*

Based on this analysis, it is more reasonable to define the RWC-clustering problem using $\mathcal{W}(\cdot, \cdot)$. Naturally, the centers in RWC-clustering problem should be clean.

## 3.1 Algorithm

The original Gonzalez's algorithm [Gonzalez, 1985] selects centers directly from the input dataset. However, in our noisy setting, selecting candidate centers directly from the noise-contaminated dataset $Q$ can lead to noisy candidate centers, which contradicts our goal of obtaining clean cluster centers. Therefore, our RWC-clustering problem requires additional mechanisms to handle noise in the candidate centers. To address this, we design a purification step to remove such noise. Then, we combine this purification step with the classical Gonzalez's algorithm [Gonzalez, 1985] to develop a tailored initialization algorithm for our RWC-clustering problem. The detailed implementation is provided in Algorithm 1.

Our Algorithm 1 takes as input a set $Q$ of probability measures and a parameter $k$, and outputs a $k$-center set $C$ consisting of $k$ probability measures. This provides a good initialization for the subsequent optimization in the post-processing stage.

---

**Algorithm 1** Seeding

1: **Input:** a set $Q = \left\{\mu^i\right\}_{i \in [m]}$ of probability measures, and a parameter $k$
2: Initialize the center set as $C = \emptyset$.
3: **for** $i = 1$ **to** $k$ **do**
4:    ▷Select candidate center $\nu$
5:    **if** $i = 1$ **then**
6:       Sample a measure $\nu$ from $Q$ uniformly at random.
7:    **else**
8:       Select the point $\nu$ that is farthest from the center set $C$ under $\mathcal{W}(\cdot, \cdot)$ according to Equation (3).
9:    **end if**
10:   ▷Purification step: purify $\nu$ to obtain $\tilde{\nu}$
11:   Perform the purification step on candidate center $\nu$, and obtain its corresponding clean center $\tilde{\nu}$ according to Equations (4) to (6).
12:      Add $\tilde{\nu}$ to center set $C$.
13: **end for**
14: **Output:** a $k$-center set $C$

---

Specifically, we select the first candidate center[2] $\nu$ from the set $Q$ uniformly at random, perform a purification step to obtain a clean center $\tilde{\nu}$, and add it to the center set $C$. For the subsequent $k - 1$ epochs, during each epoch, we select a point $\nu \in Q$ that is the farthest from the center set $C$ under $\mathcal{W}(\cdot, \cdot)$; that is, $\nu$ satisfies that

$$\nu \in \arg\max_{\nu' \in Q} \mathcal{W}(\nu', C). \tag{3}$$

Here, $\nu' \in Q$ contains noise, while the points in the center set $C$ are clean. Consequently, we adopt $\mathcal{W}(\cdot, \cdot)$ to measure the difference between them. Then, we perform the purification step on $\nu$ to obtain a clean center $\tilde{\nu}$, and add $\tilde{\nu}$ to $C$.

**Purification step:** Select the $\tau$ closest points to the candidate center $\nu$ from the set $Q$ under $\widetilde{\mathcal{W}}(\cdot, \cdot)$; that is,

$$D \in \arg\min_{D' \subseteq Q, |D'| = \tau} \sum_{\mu \in D'} \widetilde{\mathcal{W}}(\mu, \nu). \tag{4}$$

Both the candidate center $\nu$ and the data points $\mu \in Q$ contain outliers. Thus, we use $\widetilde{\mathcal{W}}(\cdot, \cdot)$ to measure the similarity between $\mu$ and $\nu$. These $\tau$ points in $D$ can induce [3] $\tau$ clean centers $\tilde{\nu}'$.

$$\widetilde{C} = \left\{ \tilde{\nu}' \mid \widetilde{\mathcal{W}}(\mu, \nu) = \mathcal{W}(\mu, \tilde{\nu}'), \mu \in D \right\} \tag{5}$$

---

[2]In our paper, the candidate center contains outliers, while the center is clean.

[3]In Equation (5), for each $\mu$, there may exist infinitely many $\tilde{\nu}'$ that satisfy the condition, but we select only one of them.

Equation (5) is computed according to [Nietert et al., 2022, Wang et al., 2024]. More specifically, we can compute the corresponding coupling matrix $\mathbf{P}$ for each $\mu \in D$. Then, we obtain $\tilde{\nu}' = \mathbf{P}^T \mathbf{1}$ , and then derive the final set $\widetilde{C}$.

Then, choose the point $\tilde{\nu} \in \widetilde{C}$ that covers all the points in $D$ with the smallest radius under $\mathcal{W}(\cdot, \cdot)$; that is,

$$\tilde{\nu} \in \arg \min_{\tilde{\nu}' \in \widetilde{C}} \max_{\mu \in D} \mathcal{W}(\mu, \tilde{\nu}'). \tag{6}$$

Then, $\tilde{\nu}$ in Equation (6) is the corresponding purified clean center of candidate center $\nu$. After the purification step, the locations of the candidate center $\nu$ and clean center $\tilde{\nu}$ remain unchanged; only the weights are adjusted.

**Intuition behind the choice for $\tau$:**  **i)** Since a candidate center $\nu$ can induce different clean centers for different $\mu \in Q$, we retain the smallest $\tau$ values of $\widetilde{\mathcal{W}}(\cdot, \nu)$ in Equation (5), rather than selecting only one. **ii)** Note that $\nu$ is a candidate center associated with a specific cluster. Points from other clusters mixed into $D$ can damage the purification of the candidate center. Moreover, the time complexity of the purification step grows quadratically with $\tau$, thus choosing a large $\tau$ can lead to significant computational overhead. Consequently, we usually set $\tau$ to be a small constant in practice.

**Remark 3.2** (Post-processing Algorithm). *Our seeding algorithm (Algorithm 1) provides a good initialization solution, but the result remains relatively coarse. To further refine the solution, we introduce a post-processing algorithm (see Algorithm 4 in the appendix), which is a combination of our purification step and the local search strategy [Lattanzi and Sohler, 2019, Choo et al., 2020].*

**Time complexity:**  Let $\mathcal{O}(\mathcal{T})$ denote the time complexity [4] of computing RWD, and $\tau$ be a constant. In Algorithm 1, selecting candidate centers during each epoch requires $\mathcal{O}(m \cdot \mathcal{T})$ time, and the purification step also takes $\mathcal{O}(m \cdot \mathcal{T})$ time. With $k$ epochs in total, the overall time complexity is $\mathcal{O}(km \cdot \mathcal{T})$. The time complexity of the post-processing algorithm is $\mathcal{O}(km \cdot \mathcal{T} + Z \cdot (km + m \cdot \mathcal{T}))$ (details are in the appendix). Therefore, the total time complexity for solving the RWC-clustering problem is $\mathcal{O}(km \cdot \mathcal{T} + Z \cdot (km + m \cdot \mathcal{T}))$.

In the context of OT, it is common to allow a constant additive error $\epsilon_+$. The OT problem can be solved in $\widetilde{\mathcal{O}}(n^2)$ time by using the existing solvers [Jambulapati et al., 2019, Cuturi, 2013], where $n$ denotes the support size of measures. Since the RWD is essentially an OT problem, the total time complexity for solving the RWC-clustering problem is $\mathcal{O}(kmn^2 + Z \cdot (km + mn^2))$.

**Remark 3.3** (Generality of the algorithmic framework of RWC-clustering problem). *Our algorithmic framework is general and can be applied to center clustering problems under other robust distances, as long as the corresponding coupling matrix can be computed efficiently. We focus on the Robust Wasserstein Distance (RWD) in particular because, under this criterion, our introduced data compression method enjoys theoretical guarantees.*

# 4   Acceleration

This section introduces a data compression technique, coreset, to accelerate computation by reducing dataset size. While coreset construction often requires an approximate solution as an anchor, the non-metric nature of RWD makes theoretical analysis difficult, preventing us from obtaining a provable approximation solution for the RWC-clustering problem. To address this, we instead compute a lower bound as the anchor, enabling coreset construction with theoretical guarantees.

**Lower bound:**  We compute a lower bound by substituting the metric in the classical Gonzalez's algorithm with $\widetilde{\mathcal{W}}(\cdot, \cdot)$ (see Algorithm 5 for details). We formalize this in the following theorem.

**Theorem 4.1** (Lower bound). *Let $\Delta$ be the optimal value of the $k$-RWC-clustering problem, i.e., $\Delta = \min_{C \subseteq \mathcal{P}(\mathcal{X}), |C|=k} cost(Q, C)$. Algorithm 5 takes set $Q$ as input and outputs a $k$-center set $C_k$ within $\mathcal{O}(km \cdot \mathcal{T})$ time. We define $\Gamma := \max_{\mu \in Q} \widetilde{\mathcal{W}}(\mu, C_k)$. Then, we have $\Gamma \leq 2\Delta$.*

---

[4]We assume that the distance between any two points in $\mathcal{X}$ can be computed within $\mathcal{O}(1)$ time.

As described in Theorem 4.1, Algorithm 5 computes a lower bound for RWC-clustering problem, which provides a theoretical guidance for subsequent coreset construction.

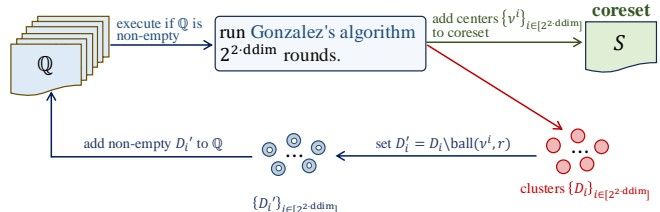

Figure 1: Coreset construction.

**Coreset:** Algorithm 2 describes a coreset construction method, which is inspired by [Ding et al., 2021, Krauthgamer and Lee, 2004, Har-Peled and Mendel, 2005, Wang et al.]. The algorithm takes as input a set $Q$ of probability measures, its doubling dimension[5] ddim, and a parameter $r$, and outputs a coreset $S$. The coreset construction relies on the Wasserstein distance, which serves as the key metric throughout the process.

Figure 1 provides an intuitive and comprehensible understanding of this method. Specifically, we begin by initializing the family $\mathbb{Q}$ of sets as $\mathbb{Q} = \{Q\}$. The following *local procedure* is then executed on every set $D \in \mathbb{Q}$ until $\mathbb{Q}$ becomes empty:

- Execute the Gonzalez's algorithm $2^{2 \cdot \text{ddim}}$ rounds on $D \in \mathbb{Q}$, yielding a set of centers $\left\{\nu^i\right\}_{i \in [2^{2 \cdot \text{ddim}}]}$ and their corresponding clusters $\{D_i\}_{i \in [2^{2 \cdot \text{ddim}}]}$. The centers are added to the coreset $S$.

- For each cluster $D_i$, we construct $D_i'$ by removing all points within a ball of radius $r$ centered at $\nu^i$, formally defined as

$$D_i' = D_i \backslash \text{ball}(\nu^i, r), \tag{7}$$

  where $\text{ball}(\nu^i, r) := \left\{\mu \mid W(\mu, \nu^i) \leq r, \mu \in D_i\right\}$.

- If $D_i'$ is non-empty, we add it to $\mathbb{Q}$. Remove the set $D$ from $\mathbb{Q}$.

---

**Algorithm 2** Coreset

---

1: **Input:** a set $Q = \left\{\mu^i\right\}_{i \in [m]}$ of probability measures, doubling dimension ddim and parameter $r$

2: Initialize $\mathbb{Q} = \{Q\}$ and $S = \emptyset$.
3: **for** set $D$ **in** $\mathbb{Q}$ **do**
4:     ▷local procedure
5:     Run Gonzalez's algorithm $2^{2 \cdot \text{ddim}}$ rounds on $D$, yielding centers $\left\{\nu^i\right\}_{i \in [2^{2 \cdot \text{ddim}}]}$ and clusters $\{D_i\}_{i \in [2^{2 \cdot \text{ddim}}]}$.
6:     Set $S = S \cup \left\{\nu^i\right\}_{i \in [2^{2 \cdot \text{ddim}}]}$.
7:     Construct $D_i'$ by removing points within a ball of radius $r$ centered at $\nu^i$ according to Equation (7).
8:     Add all non-empty $D_i'$ to $\mathbb{Q}$; i.e., $\mathbb{Q} = \mathbb{Q} \cup \{D_i'\}$.
9:     Set $\mathbb{Q} = \mathbb{Q} \backslash \{D\}$.
10: **end for**
11: **Input:** coreset $S$

---

**Theorem 4.2** (Coreset property). *Let* ddim *be the doubling dimension of $Q$ and $R$ be the radius of $Q$ under Wasserstein distance, i.e., $W(\mu, \nu) \leq 2R$ for any $\mu, \nu \in Q$. We set $r = \mathcal{O}(\epsilon\Gamma)$, then Algorithm 2 outputs an $\epsilon$-coreset $S$ with $|S| = \mathcal{O}((\frac{R}{r})^{2 \cdot \text{ddim}})$ for $k$-RWC-clustering problem on $Q$ within $\mathcal{O}(2^{2 \cdot \text{ddim}} \cdot |Q| \cdot \mathcal{T} \cdot \log \frac{R}{r})$ time.*

---

[5]Given a metric space $(Q, W)$, its doubling dimension [Huang et al., 2018, Wang et al., 2024] is defined as the smallest integer ddim such that any ball with radius $2r$ can be covered by at most $2^{\text{ddim}}$ balls of radius $r$.

**Corollary 4.3.** *Algorithm 2 takes $Q$ as its input and outputs the coreset $S$. The output $S$ satisfies the following property*

$$\left| \min_{C \in \mathcal{P}(\mathcal{X}), |C|=k} \textsf{cost}(Q, C) - \min_{C' \in \mathcal{P}(\mathcal{X}), |C'|=k} \textsf{cost}(S, C') \right| \leq \min_{C \in \mathcal{P}(\mathcal{X}), |C|=k} \epsilon \cdot \textsf{cost}(Q, C).$$

**A good proxy:**  As stated in Theorem 4.2, for any subset $C \subseteq \mathcal{P}(\mathcal{X})$ with $|C| = k$, the values computed on the coreset can closely approximate the values on the original dataset within an $\epsilon$-relative error. That is,

$$\textsf{cost}(S, C) \approx \textsf{cost}(Q, C). \tag{8}$$

Furthermore, according to Corollary 4.3, the optimal value computed on the coreset is approximately the same as the optimal value computed on the original dataset. That is,

$$\min_{C \subseteq Q, |C|=k} \textsf{cost}(S, C) \approx \min_{C \subseteq Q, |C|=k} \textsf{cost}(Q, C). \tag{9}$$

According to Equations (8) and (9), we have that the coreset $S$ serves as a good proxy of the original dataset $Q$ for the RWC-clustering problem.

**Remark 4.4** (Enhancing scalability for coreset construction)**.** *We can accelerate coreset construction by leveraging the merge-and-reduce framework [Bentley and Saxe, 1980, Har-Peled and Mazumdar, 2004], which is efficient in both computation and communication. Specifically, a large dataset can be partitioned into smaller subsets and distributed across multiple machines for parallel computation, which significantly improves time efficiency. Furthermore, since the coreset is a subset of the original dataset, only the indices of the data points, rather than the data items themselves, need to be transmitted during machine synchronization. This makes the communication overhead negligible, ensuring excellent scalability of our coreset approach. Additionally, the merge-and-reduce framework enables our approach to adapt seamlessly to streaming data, making it highly effective and efficient in dynamic data processing scenarios.*

**Remark 4.5.** *In the process of constructing the coreset, Wasserstein distance is employed primarily to ensure theoretical rigor. In practice, the construction of the coreset can also utilize $\mathcal{W}(\cdot, \cdot)$.*

## 5   Experiments

This section demonstrates the effectiveness of our RWC-clustering problem and the efficiency of the coreset method. All the experiments were performed on a server with an AMD EPYC 9754 128-Core Processor with 18 vCPUs, 60GB of RAM, and Python 3.12. The server utilized an RTX 4090D GPU with 24GB of VRAM. We used the POT library [Flamary et al., 2021] to compute the Wasserstein distance (WD) [Bonneel et al., 2011] and unbalanced optimal transport (UOT) [Chizat et al., 2018, Frogner et al., 2015]. The reported results are averaged over five runs.

Due to space constraints, we report experimental results only on **ModelNet10** [Wu et al., 2015] dataset in the main text. More experiments on other datasets, including **Geometric shapes**, **MNIST** [LeCun et al., 1998], **KITTI** [Geiger et al., 2012], **ShapeNetCore** [Chang et al., 2015], **ScanObjectNN** [Uy et al., 2019], **nuScenes Mini** [Caesar et al., 2020], as well as related ablation studies, are in appendix.

**ModelNet10** [Wu et al., 2015] is a standard dataset containing 3D CAD models. Each CAD model is discretized and represented as a point set. We extract object-level point cloud instances from the dataset and uniformly sample 300 points from each to construct our point cloud dataset.

For the point set $\{x_i\}_{i \in [n]}$ corresponding to a specific data item in the above datasets, we represent it as a probability measure $\mu = \sum_{i=1}^{n} \frac{1}{n} \delta_{x_i}$ to construct the clean dataset $Q^0$. The noisy dataset $Q$ is generated by adding clustered noise with $\zeta$ mass following a Gaussian distribution $\mathcal{N}(\mu, \sigma^2)$ to each clean probability measure.

To evaluate the performance of our methods, we consider the following three criteria: i) Runtime: This includes the sampling time and the clustering time required to compute the $k$-center set $C$. ii) cost-cd: Defined as $\max_{\mu \in Q^0} \min_{\nu \in C} W(\mu, \nu)$, it quantifies the distance between the center set $C$ and the original clean dataset $Q^0$. iii) cost-nd: Defined as $\max_{\mu \in Q} \min_{\nu \in C} \mathcal{W}(\mu, \nu)$, it evaluates the distance from center set $C$ to the noisy dataset $Q$. The baselines for these three criteria are established using the results computed on the original full dataset for comparison.

**Effectiveness of our RWC-clustering problem:** To demonstrate the effectiveness of our RWC-clustering problem, a series of experiments were conducted on ModelNet10 dataset. We randomly sample 200 data items from the ModelNet10 to form the dataset. We compare the clustering quality computed by UOT-based clustering, WD-based clustering, and RWC-clustering. All three methods follow the same framework, where seeding is used for initialization, followed by a local search refinement. Specifically, the UOT-based clustering algorithm is derived by replacing RWD with UOT in Algorithms 1 and 4. Since WD is a metric, thus the WD-based clustering can directly use the existing Gonzalez's algorithm [Gonzalez, 1985] along with the local search method [Lattanzi and Sohler, 2019, Choo et al., 2020].

As shown in Table 1, the WD-based clustering exhibits the worst performance. The UOT-based clustering outperforms the WD-based clustering; however, the inclusion of an entropy regularization term causes a diffusion effect that hinders noise removal, leaving some residual noise inevitably. In contrast, our RWC-clustering approach achieves the best denoising performance, outperforming the other two methods.

Table 1: Comparison of our RWC-clustering with UOT-based clustering and WD-based clustering on ModelNet10 dataset. We fix the dataset size as 200, $k = 10$, $Z = 200$, $\sigma = 1$, $\tau = 10$ and $\zeta = 0.1$, where $Z$ is the number of iterations in post-processing procedure (i.e., Algorithm 4 in appendix).

| Dataset | Method | cost-nd($\downarrow$) | cost-cd($\downarrow$) | Runtime($\downarrow$) |
|---|---|---|---|---|
| | RWC-clustering | $\mathbf{2.20}_{\pm 0.13}$ | $\mathbf{2.40}_{\pm 0.05}$ | $475.98_{\pm 2.39}$ |
| ModelNet10 | UOT-based clustering | $4.07_{\pm 0.30}$ | $4.26_{\pm 0.40}$ | $446.11_{\pm 7.66}$ |
| | WD-based clustering | $5.32_{\pm 0.48}$ | $7.25_{\pm 0.72}$ | $475.98_{\pm 2.39}$ |

**Efficiency of our coreset method:** We show the efficiency of our coreset (CS) method by comparing it against the uniform sampling (US) method. To ensure fairness, both sampling methods (SM) employed the same sampling size (SS). The green dashed line indicates the results on the full dataset, which serves as the baseline for comparison. Figure 2 illustrates the performance of our CS method on ModelNet10 dataset. Although our CS method is slightly more time-consuming compared to the US method, it remains significantly more efficient than processing the original dataset. Moreover, on both cost-cd and cost-nd criteria, our CS method consistently outperforms the US method.

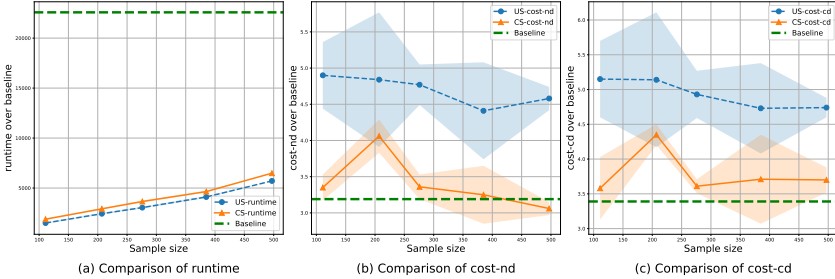

| (a) Comparison of runtime | (b) Comparison of cost-nd | (c) Comparison of cost-cd |

Figure 2: Comparison of the US method and our CS method across varying sample sizes on ModelNet10 dataset. We fix $k = 10$, $Z = 200$, $\sigma = 1$, $\tau = 10$ and $\zeta = 0.1$.

Due to the randomness inherent in our algorithms and the approximate nature of the derived $k$-center set, some fluctuations are inevitable.

## 6 Conclusion and future work

In this paper, we introduce the **R**obust **W**asserstein **C**enter clustering (RWC-clustering) problem, and propose an efficient algorithm to solve it. Additionally, we introduce a coreset method to accelerate computations by compressing the dataset; moreover, we provide new analysis to establish theoretical guarantees for the coreset method under the RWC-clustering setting. For future work, we will explore the corresponding *robust Wasserstein k-means clustering* problem, along with its approximation algorithms and coreset techniques.

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

**Limitations:** Since the robust Wasserstein distance is not a true metric and does not satisfy the triangle inequality, theoretical analysis becomes challenging, and no approximation algorithm with provable guarantees has been obtained.

**Broader impact:** This study contributes to the representation of complex data in noisy environments. Currently, we are not aware of any potential negative consequences arising from this work.

# A   Other preliminaries

**Definition A.1** (Optimal transport (OT) [Peyré et al., 2017]). *Let* $\mu = \sum_{i=1}^{n} a_i \delta_{x_i}$ *and* $\nu = \sum_{j=1}^{n} b_j \delta_{y_j}$ *be two probability measures with weights* $\mathbf{a}, \mathbf{b} \in \mathbb{R}_+^n$, *respectively. Given a cost matrix* $\mathbf{D} \in \mathbb{R}_+^{n \times n}$, *the OT distance between* $\mu$ *and* $\nu$ *is*

$$\mathcal{OT}(\mu, \nu) := \min_{\mathbf{P} \in \Pi(\mathbf{a}, \mathbf{b})} \langle \mathbf{P}, \mathbf{D} \rangle,$$

*where* $\Pi(\mathbf{a}, \mathbf{b}) := \left\{ \mathbf{P} \in \mathbb{R}_+^{n \times n} \mid \mathbf{P}\mathbf{1} = \mathbf{a}, \mathbf{P}^T\mathbf{1} = \mathbf{b} \right\}$ *is the set of all feasible couplings and* $\mathbf{1}$ *is the vector of all ones.*

The Wasserstein distance is a special case of OT. The cost matrix $\mathbf{D}$ of Wasserstein distance must be induced by a distance function, while the cost matrix of OT only needs to be induced by a positive function. Thus, the Wasserstein distance is a metric on $\mathcal{P}(\mathcal{X})$, whereas OT is not a metric.

The doubling dimension provides a means for describing the growth rate of the data set with respect to certain metric.

**Definition A.2** (Doubling dimension [Huang et al., 2018, Wang et al., 2024]). *Let* $(Q, W)$ *be a metric space, where* $W(\cdot, \cdot)$ *is a metric on* $Q$. *The doubling dimension of* $(Q, W)$ *is the smallest integer **ddim** such that every ball of radius* $2r$ *can be covered by at most* $2^{\mathsf{ddim}}$ *balls of radius* $r$.

In real-world datasets, data often exhibit inherent regularities, leading to a relatively low intrinsic dimension. Therefore, assuming a low doubling dimension is usually reasonable.

Moreover, in practical applications, we do not need to know the exact value of the doubling dimension in advance. We usually start with a small value and adjust as needed. In our experiments, we assume the doubling dimension to be 1. The lack of precise knowledge of the doubling dimension does not affect practical applications.

**Definition A.3** ($r$-cover[Shalev-Shwartz and Ben-David, 2014, Mohri, 2018]). *Given a metric space* $(\mathcal{X}, \mathrm{dist})$, *a set* $A \subseteq \mathcal{X}$ *is an* $r$-cover *of* $Q \subseteq \mathcal{X}$, *if for any* $x \in Q$, *there exists* $x' \in A$ *satisfying* $\mathrm{dist}(x, x') \leq r$.

Note that, a cover of a set does not need to be a subset of it.

**Gonzalez's algorithm** [Gonzalez, 1985], a 2-approximation algorithm for the metric $k$-center problem, is often used as an initialization method in clustering tasks.

Let $Q$ be a data set and $\mathrm{dist}(\cdot, \cdot)$ be the metric on $Q$. We iteratively select the point farthest from the currently chosen centers as the new center. The sequential nature of center selection leads to dependencies between steps, which poses challenges for parallelization. It takes $\mathcal{O}(mk)$ time, where $m$ is the size of the dataset, and $k$ represents the number of centers. When $k$ or $m$ is large, the computational complexity becomes a bottleneck. The details are in Algorithm 3.

---
**Algorithm 3** Gonzalez's algorithm

---
**Input:** data set $Q$, and a parameter $k$
Sample $c \in Q$ uniformly at random, and set $C_1 = \{c\}$.
**for** $i = 2$ **to** $k$ **do**
    Select $c \in Q$ that is farthest from the center set $C_{i-1}$.
**end for**
Set $C = C_k$.
**Output:** a $k$-center set $C$

---

# B Post-processing algorithm

---

**Algorithm 4** Post-processing

---
1: **Input:** a set $Q = \left\{\mu^i\right\}_{i \in [m]}$ of probability measures, and a $k$-center set $C$
2: **for** $i = 1$ **to** $Z$ **do**
3:    ▷Sampling
4:    Sample $\nu \in Q$ with probability $\frac{\mathcal{W}(\nu, C)}{\sum_{\nu' \in Q} \mathcal{W}(\nu', C)}$.
5:    ▷Purification
6:    Purify $\nu$ into $\tilde{\nu}$ according to Equations (4) to (6).
7:    ▷Swapping
8:    **if** $\exists \nu' \in C$, s.t., $\mathsf{cost}(Q, C \setminus \{\nu'\} \cup \{\tilde{\nu}\}) < \mathsf{cost}(Q, C)$ **then**
9:       $C = C \setminus \{\nu'\} \cup \{\tilde{\nu}\}$.
10:    **end if**
11: **end for**
12: **Output:** a $k$-center set $C$

---

**Post-processing algorithm:** Algorithm 4 is inspired by the local search algorithm [Lattanzi and Sohler, 2019, Choo et al., 2020], and serves as a post-processing procedure for Algorithm 1 to further refine the solution. The input is a set $Q$ of probability measures and an initialized solution (i.e., $k$-center set), while the output is the refined solution.

The solution $C$ is refined over $Z$ epochs, with each epoch consisting of three steps: sampling, purification, and swapping. Specifically, in each epoch, we sample a candidate center $\nu \in Q$ according to a probability proportional to its cost, i.e., $\frac{\mathcal{W}(\nu, C)}{\sum_{\nu' \in Q} \mathcal{W}(\nu', C)}$. Then, we apply the purification step in Algorithm 1 to purify the candidate center $\nu$ into a clean center $\tilde{\nu}$. Next, if there exists a center $\nu' \in C$ such that replacing $\nu'$ with $\tilde{\nu}$ results in a reduction of the cost, we replace $\nu'$ with $\tilde{\nu}$.

**Time complexity:** Let $\mathcal{O}(\mathcal{T})$ denote the time complexity[6] of computing RWD, and $\tau$ be a constant. In Algorithm 1, selecting candidate centers during each epoch requires $\mathcal{O}(m \cdot \mathcal{T})$ time, and the purification step also takes $\mathcal{O}(m \cdot \mathcal{T})$ time. With $k$ epochs in total, the overall time complexity is $\mathcal{O}(km \cdot \mathcal{T})$.

For Algorithm 4, initializing the distance matrix between $C$ and $Q$ takes $\mathcal{O}(km \cdot \mathcal{T})$ time. During each epoch, the operations require $\mathcal{O}(km + m \cdot \mathcal{T})$ time. Assuming $Z$ epochs in total, the total time complexity is $\mathcal{O}(km \cdot \mathcal{T} + Z \cdot (km + m \cdot \mathcal{T}))$.

**Remark B.1.** *To ensure the fairness of experimental comparisons, we have fixed $Z$ in our paper. In practical applications, we can indeed design early stopping criteria based on specific needs. For instance, stopping when no improvement occurs is a optional strategy.*

# C Omitted proofs and details

## C.1 Intuition of RWC-clustering problem

Let $Q = \left\{\mu^i\right\}_{i \in [m]}$ be a set of probability measures. We define the ball center at $\nu$ with radius $r$ under metric $\mathcal{W}(\cdot, \cdot)$ as

$$\mathsf{ball}_{\mathcal{W}}(\nu, r) := \{\mu \mid \mathcal{W}(\mu, \nu) \leq r, \mu \in Q\}. \tag{10}$$

**Lemma C.1.** *If $\mu^0, \mu^1 \in \mathcal{P}(\mathcal{X})$ are in $\mathsf{ball}_{\mathcal{W}}(\nu, r)$, we have $\widetilde{\mathcal{W}}(\mu^0, \mu^1) \leq 2r$.*

---

[6]We assume that the distance between any two points in $\mathcal{X}$ can be computed within $\mathcal{O}(1)$ time.

**Proof of Lemma C.1.** Since $\mu^0, \mu^1$ are in $\mathsf{ball}_{\mathcal{W}}(\nu, r)$, we have $\mathcal{W}(\mu^0, \nu) \leq r, \quad \mathcal{W}(\mu^1, \nu) \leq r$. Assume that $\hat{\mu}^0, \hat{\mu}^1$ are the clean probability measures induced by $\mu^0, \mu^1$, respectively. That is,

$$\mathcal{W}(\mu^0, \nu) = W(\hat{\mu}^0, \nu), \quad \mathcal{W}(\mu^1, \nu) = W(\hat{\mu}^1, \nu). \tag{11}$$

According to Definition 2.1, we have $\widetilde{\mathcal{W}}(\mu^0, \mu^1) \leq \mathcal{W}(\mu^0, \hat{\mu}^1) \leq W(\hat{\mu}^0, \hat{\mu}^1)$. Then, by combining triangle inequality property of Wasserstein distance with Equation (11), we obtain

$$\widetilde{\mathcal{W}}(\mu^0, \mu^1) \leq W(\hat{\mu}^0, \hat{\mu}^1) \leq W(\hat{\mu}^0, \nu) + W(\hat{\mu}^1, \nu) = \mathcal{W}(\mu^0, \nu) + \mathcal{W}(\mu^1, \nu) \leq 2r.$$

$\square$

Lemma C.1 implies that, under $\mathcal{W}(\cdot, \cdot)$, the difference between two data points located within a small ball is relatively small.

## C.2 Lower bound

Algorithm 5 essentially replaces the metric in the classical Gonzalez's algorithm with $\widetilde{\mathcal{W}}(\cdot, \cdot)$. Specifically, let $C_i$ be the center set containing $i$ centers. We initially select a point $\mu$ randomly from the input dataset $Q$, and initialize the center set as $C_1 = \{\mu\}$. Then, for the $i$-th epoch with $2 \leq i \leq k$, we choose the point $\mu \in Q$ that is farthest from the previous center set $C_{i-1}$, and set $C_i = C_{i-1} \cup \{\mu\}$; formally, the center $\mu$ selected, except in the first epoch, satisfies that

$$\mu \in \arg\max_{\mu' \in Q} \widetilde{\mathcal{W}}(\mu', C_{i-1}), \tag{12}$$

where $\widetilde{\mathcal{W}}(\mu, C_{i-1}) := \min_{\nu \in C_{i-1}} \widetilde{\mathcal{W}}(\mu, \nu)$. Notably, no purification step is applied during this process, thus the centers in $C_i$ for $i \in [k]$ contains outliers.

As described in Theorem 4.1, Algorithm 5 computes a lower bound for RWC-clustering problem, which provides a theoretical guidance for subsequent coreset construction.

---

**Algorithm 5** Lower bound

---

1: **Input:** a set $Q = \{\mu^i\}_{i \in [m]}$ of probability measures, and a parameter $k$
2: Sample $\mu \in Q$ uniformly at random, and set $C_1 = \{\mu\}$.
3: **for** $i = 2$ **to** $k$ **do**
4:     Select $\mu \in Q$ that is farthest from the center set $C_{i-1}$ under $\widetilde{\mathcal{W}}(\cdot, \cdot)$ according to Equation (12).
5: **end for**
6: **Output:** a $k$-center set $C_k$

---

**Theorem 4.1** (Lower bound). *Let $\Delta$ be the optimal value of the $k$-RWC-clustering problem, i.e., $\Delta = \min_{C \subseteq \mathcal{P}(\mathcal{X}), |C| = k} \mathsf{cost}(Q, C)$. Algorithm 5 takes set $Q$ as input and outputs a $k$-center set $C_k$ within $\mathcal{O}(km \cdot \mathcal{T})$ time. We define $\Gamma := \max_{\mu \in Q} \widetilde{\mathcal{W}}(\mu, C_k)$. Then, we have $\Gamma \leq 2\Delta$.*

Let $C^* = \{\nu_*^i\}_{i \in [k]}$ be the optimal solution to the RWC-clustering problem, and $\Delta$ be its corresponding optimal value; that is,

$$\min_{C \subseteq \mathcal{P}(\mathcal{X}), |C| = k} \mathsf{cost}(Q, C) = \mathsf{cost}(Q, C^*) = \Delta. \tag{13}$$

Let $Q_i^*, i \in [k]$ be the clusters induced by $\nu_*^i, i \in [k]$, where each point is assigned to the nearest cluster center. That is,

$$\mathcal{W}(\mu, \nu_*^i) = \mathcal{W}(\mu, C^*) \leq \Delta \quad \text{for } \mu \in Q_i^*. \tag{14}$$

The set $C_k$ is the output of Algorithm 5. The centers in $C_k$ contain outliers, whereas the centers in $C^*$ are clean.

**Lemma C.2.** *For any $\mu \in Q_i^*, i \in [k]$, we have $\widetilde{\mathcal{W}}(\mu, C_k) \leq 2\Delta$ if $|Q_i^* \cap C_k| \geq 1$.*

549 **_Proof of Lemma C.2_**. Let $\nu \in Q_i^* \cap C_k$. Since $\nu_*^i$ is the nearest center of $\mu \in Q_i^*$, by using
550 Equation (13), we have

$$\mathcal{W}(\mu, \nu_*^i) \leq \Delta \text{ for all } \mu \in Q_i^*. \tag{15}$$

551 We know that both $\mu, \nu$ are in cluster $Q_i^*$, thus $\mu, \nu \in \mathsf{ball}_{\mathcal{W}}(\nu_*^i, \Delta)$. Then, by using Lemma C.1,
552 we obtain $\widetilde{\mathcal{W}}(\mu, \nu) \leq 2 \cdot \Delta$. According to the definition $\widetilde{\mathcal{W}}(\mu, C_k) := \min_{\nu' \in C_k} \widetilde{\mathcal{W}}(\mu, \nu')$, we have
553 $\widetilde{\mathcal{W}}(\mu, C_k) \leq \widetilde{\mathcal{W}}(\mu, \nu)$. Till now, we obtain $\widetilde{\mathcal{W}}(\mu, C_k) \leq 2\Delta$.

554 □

555 **_Proof of Theorem 4.1_**. In order to prove the conclusion, we will discuss the proof in two cases,
556 which is inspired by [Gonzalez, 1985].

557 **Case 1:** Each $Q_i^*$ contains exactly one center $\nu \in C_k$.
558

According to Lemma C.2, we have $\widetilde{\mathcal{W}}(\mu, C_k) \leq 2\Delta$ for all $\mu \in Q_i^*, i \in [k]$. Since the collection
$\{Q_i^*\}_{i \in [k]}$ forms a partition of $Q$, we have

$$Q = \sqcup_{i=1}^k Q_i^*.$$

559 Thus, it follows that $\widetilde{\mathcal{W}}(\mu, C_k) \leq 2\Delta$ for all $\mu \in Q$. That is, $\Gamma \leq 2\Delta$ holds.

560 **Case 2:** Some $Q_i^*$ contains multiple centers, i.e., $|C_k \cap Q_i^*| \geq 2$.
561

562 Without loss of generality, suppose that $C_{i_0}$ is the first center set such that $|C_{i_0} \cap Q_i^*| = 2$ for some
563 $i \in [k]$, with $C_{i_0} \cap Q_i^* = \{\nu, \nu^{i_0}\}$, where $\nu^{i_0}$ is the last center added to $C_{i_0}$.
564 From the Line 3 of Algorithm 5, we know that

$$\widetilde{\mathcal{W}}(\mu, C_{i_0-1}) \leq \widetilde{\mathcal{W}}(\nu^{i_0}, C_{i_0-1}) \quad \text{for all } \mu \in Q.$$

565 Since $\nu \in C_{i_0-1}$, we have

$$\widetilde{\mathcal{W}}(\nu^{i_0}, C_{i_0-1}) \leq \widetilde{\mathcal{W}}(\nu^{i_0}, \nu) \quad \text{for } i \in [k-1].$$

566 We know that both $\nu, \nu^{i_0}$ are in cluster $Q_i^*$, thus $\nu, \nu^{i_0} \in \mathsf{ball}_{\mathcal{W}}(\nu_*^i, \Delta)$. Then, by using Lemma C.1,
567 we obtain $\widetilde{\mathcal{W}}(\nu, \nu^{i_0}) \leq 2 \cdot \Delta$.
568 Till now, we achieve that

$$\widetilde{\mathcal{W}}(\mu, C_{i_0-1}) \leq 2\Delta \quad \text{for all } \mu \in Q.$$

569 From Line 3 of Algorithm 5, it follows that

$$\widetilde{\mathcal{W}}(\mu, C_k) \leq \widetilde{\mathcal{W}}(\mu, C_{i_0-1}) \quad \text{for all } \mu \in Q.$$

570 Therefore, we obtain

$$\Gamma \leq 2\Delta \quad \text{for Case 2}.$$

571 In conclusion, for both Case 1 and Case 2, the inequality $\Gamma \leq 2\Delta$ holds.

572 **Time complexity:** The time complexity for selecting each center is $\mathcal{O}(m \cdot \mathcal{T})$. Since we need to
573 select $k$ centers in total, the overall time complexity is $\mathcal{O}(km\mathcal{T})$.

574 □

## C.3 Analysis of coreset (Theorem 4.2)

**Theorem 4.2** (Coreset property). *Let ddim be the doubling dimension of $Q$ and $R$ be the radius of $Q$ under Wasserstein distance, i.e., $W(\mu, \nu) \leq 2R$ for any $\mu, \nu \in Q$. We set $r = \mathcal{O}(\epsilon\Gamma)$, then Algorithm 2 outputs an $\epsilon$-coreset $S$ with $|S| = \mathcal{O}((\frac{R}{r})^{2 \cdot ddim})$ for $k$-RWC-clustering problem on $Q$ within $\mathcal{O}(2^{2 \cdot ddim} \cdot |Q| \cdot \mathcal{T} \cdot \log \frac{R}{r})$ time.*

We introduce two sets, $\mathsf{Set}(\mu)$ and $\mathsf{Set}(\xi)$, defined as follows

$$\mathsf{Set}(\mu) = \left\{ \mu'' = \frac{\mu'}{1-\zeta} \in \mathcal{P}(\mathcal{X}) \mid \mu' \leq \mu, \|\mu' - \mu\|_{\mathrm{TV}} = \zeta \right\}$$

and

$$\mathsf{Set}(\xi) = \left\{ \xi'' = \frac{\xi'}{1-\zeta} \in \mathcal{P}(\mathcal{X}) \mid \xi' \leq \xi, \|\xi' - \xi\|_{\mathrm{TV}} = \zeta \right\},$$

where $\mathsf{Set}(\mu)$ and $\mathsf{Set}(\xi)$ represent the sets of feasible clean probability measures with $\zeta$ mass of outliers removed from $\mu, \nu$ according to $\mathcal{W}(\mu, \cdot)$ and $\mathcal{W}(\xi, \cdot)$, respectively.

Roughly speaking, the following theorem illustrates that if two probability measures $\mu$ and $\xi$ are similar, then the sets of feasible clean probability measures they induce, denoted by $\mathsf{Set}(\mu)$ and $\mathsf{Set}(\xi)$, respectively, are also similar, i.e., $\mathsf{Set}(\mu) \approx \mathsf{Set}(\xi)$. Specifically, for any $\mu''$ in $\mathsf{Set}(\mu)$, there exists a corresponding $\xi''$ in $\mathsf{Set}(\xi)$ that is close to $\mu''$ under Wasserstein distance. Conversely, for every $\xi''$ in $\mathsf{Set}(\xi)$, there exists a $\mu'' \in \mathsf{Set}(\mu)$ that is close to $\zeta''$. Consequently, the sets $\mathsf{Set}(\mu)$ and $\mathsf{Set}(\xi)$ are $r$-cover of each other.

**Lemma C.3.** *Given any $W(\mu, \xi) \leq r$, the sets $\mathsf{Set}(\mu)$ and $\mathsf{Set}(\xi)$ are $\frac{r}{1-\zeta}$-cover of each other.*

***Proof of Lemma C.3.*** Without loss of generality, let $\mu = \sum_{i=1}^{n} a_i \delta_{x_i}$ and $\xi = \sum_{j=1}^{n} b_j \delta_{y_j}$. Let $\mathbf{P}^*$ denote the optimal coupling induced by $W(\mu, \xi)$; that is,

$$W(\mu, \xi) = \langle \mathbf{P}^*, \mathbf{D} \rangle,$$

where $\mathbf{D}$ is the cost matrix between $\mu$ and $\xi$.

For any $\mu'' \in \mathsf{Set}(\mu)$, we have $\mu' = (1-\zeta) \cdot \mu'' = \sum_{i=1}^{n} a_i' \delta_{x_i}$ with $\mathbf{a}'$ being its weights. We can construct a $\mathbf{P}'$ satisfying

$$\mathbf{P}'\mathbf{1} = \mathbf{a}', \quad \mathbf{P}'\mathbf{1}^T \leq \mathbf{b}, \quad \mathbf{P}' \leq \mathbf{P}^*, \quad \mathbf{P}' \in \mathbb{R}^{n \times n}.$$

Let $\mathbf{b}' = \mathbf{P}'\mathbf{1}^T$, and construct $\xi' = \sum_{j=1}^{n} b_j' \delta_{y_j}$. Transferring $\mu'$ to $\xi'$ according to the flow matrix $\mathbf{P}'$, we achieve $\langle \mathbf{P}', \mathbf{D} \rangle \leq \langle \mathbf{P}^*, \mathbf{D} \rangle = W(\mu, \xi) \leq r$. Clearly, $\frac{\mathbf{P}'}{1-\zeta}$ is a feasible flow for $\mu'' = \frac{\mu'}{1-\zeta}$ and $\xi'' = \frac{\xi'}{1-\zeta}$ under the Wasserstein distance. Therefore, $W(\mu'', \xi'') \leq \langle \frac{\mathbf{P}'}{1-\zeta}, \mathbf{D} \rangle \leq \frac{r}{1-\zeta}$.

From the above, we can find a measure $\mu'' = \frac{\mu'}{1-\zeta} \in \mathsf{Set}(\mu)$ such that $W\left(\frac{\mu'}{1-\zeta}, \frac{\xi'}{1-\zeta}\right) \leq \frac{r}{1-\zeta}$ holds for any $\xi'' = \frac{\xi'}{1-\zeta} \in \mathsf{Set}(\xi)$. Similarly, we can find a measure $\xi'' \in \mathsf{Set}(\xi)$ such that $W\left(\frac{\mu'}{1-\zeta}, \frac{\xi'}{1-\zeta}\right) \leq \frac{r}{1-\zeta}$ holds for any $\mu'' \in \mathsf{Set}(\mu)$.

Thus, we have demonstrated that the sets $\mathsf{Set}(\mu)$ and $\mathsf{Set}(\xi)$ are $\frac{r}{1-\zeta}$-covers of each other.

$\square$

Let $g : \mathcal{P}(\mathcal{X}) \to \mathbb{R}, a \mapsto g(a)$ be a function. The following lemma illustrates that if for any $a \in A$, there exists $b \in B$ such that $g(a) \approx g(b)$, and vice vise; then, the minimum value of $g(\cdot)$ over the two sets $A$ and $B$ is close.

**Lemma C.4.** *If for every $a \in A$, there exists $b \in B$ such that $g(a) \in g(b) \pm r$, and vice versa, then the minimum values of $g$ over sets $A$ and $B$ are approximately equal; that is,*

$$\left| \min_{a' \in A} g(a') - \min_{b' \in B} g(b') \right| \leq r.$$

609   ***Proof of Lemma C.4.*** From the given conditions, for any $a \in A$, there exists $b \in B$ such that

$$g(b) - r \leq g(a) \leq g(b) + r. \tag{16}$$

610   Similarly, for any $b \in B$, there exists $a \in A$ such that

$$g(a) - r \leq g(b) \leq g(a) + r. \tag{17}$$

611   Let $b^*$ be the point where $g(\cdot)$ achieves its minimum on $B$, i.e.,

$$g(b^*) = \min_{b' \in B} g(b').$$

612   According to Equation (17), there exists $a' \in A$ such that

$$g(a') - r \leq g(b^*) \leq g(a') + r.$$

613   This implies that

$$g(a') - g(b^*) \leq r.$$

614   Then, we have

$$\min_{a' \in A} g(a') - \min_{b' \in B} g(b') = \min_{a' \in A} g(a') - g(b^*) \leq r. \tag{18}$$

615   Similarly, by using Equation (16), we also obtain

$$\min_{b' \in B} g(b') - \min_{a' \in A} g(a') \leq r. \tag{19}$$

616   By combining Equations (18) and (19), it follows that

$$\left| \min_{a' \in A} g(a') - \min_{b' \in B} g(b') \right| \leq r.$$

617   $\square$

618   The following theorem illustrates that if two functions are approximately equal pointwise, then their
619   minimum values are also approximately the same.

620   **Lemma C.5.** *Given two functions*

$$g, f : \mathcal{P}(\mathcal{X}) \to \mathbb{R}, \tag{20}$$

621   *if* $|g(\mu) - f(\mu)| \leq r$ *for all* $\mu \in Q$, *then we have* $|\min_{\mu \in Q} g(\mu) - \min_{\mu' \in Q} f(\mu')| \leq r$.

622   ***Proof of Lemma C.5.*** Assume that $\min_{\mu \in Q} g(\mu) > \min_{\mu' \in Q} f(\mu')$. Let $f$ achieve its minimum at
623   $\mu^*$, that is, $\min_{\mu' \in Q} f(\mu') = f(\mu^*)$. Then, we have

$$\left| \min_{\mu \in Q} g(\mu) - \min_{\mu' \in Q} f(\mu') \right| = \min_{\mu \in Q} g(\mu) - \min_{\mu' \in Q} f(\mu') = \min_{\mu \in Q} g(\mu) - f(\mu^*).$$

624   According to Equation (20), there must exist $g(\mu^*) - f(\mu^*) \leq r$, implying $\min_{\mu \in Q} g(\mu) - f(\mu^*) \leq r$.
625   Consequently, $\min_{\mu \in Q} g(\mu) - \min_{\mu' \in Q} f(\mu') \leq r$ holds. Similarly, for the case
626   $\min_{\mu \in Q} g(\mu) \leq \min_{\mu' \in Q} f(\mu')$, we can also derive $\min_{\mu \in Q} f(\mu) - \min_{\mu' \in Q} g(\mu') \leq r$.
627

628   Till now, we obtain the final conclusion.   $\square$

629   ***Proof of Theorem 4.2.*** For any center $\nu \in C \subseteq \mathcal{P}(\mathcal{X})$ and $\mu'' \in \mathsf{Set}(\mu)$, there exists $\xi'' \in \mathsf{Set}(\xi)$
630   such that $W(\mu'', \xi'') \leq \frac{r}{1-\zeta}$ according to Lemma C.3.

By using the triangle inequality, we have

$$W(\xi'', \nu) - W(\mu'', \xi'') \leq W(\mu'', \nu) \leq W(\xi'', \nu) + W(\mu'', \xi''). \tag{21}$$

This leads to the following inequality

$$|W(\mu'', \nu) - W(\xi'', \nu)| \leq W(\mu'', \xi'') \leq \frac{r}{1-\zeta}. \tag{22}$$

The above result shows that for any $\mu'' \in \mathsf{Set}(\mu)$, there exists $\xi'' \in \mathsf{Set}(\xi)$ such that $W(\mu'', \nu) \in W(\xi'', \nu) \pm \frac{r}{1-\zeta}$.

Similarly, for any $\xi'' \in \mathsf{Set}(\xi)$, we also have $W(\xi'', \nu) \in W(\mu'', \nu) \pm \frac{r}{1-\zeta}$.

Let $g(\cdot) := W(\cdot, \nu)$. By applying Lemma C.4, we can deduce

$$\left| \min_{\mu'' \in \mathsf{Set}(\mu)} W(\mu'', \nu) - \min_{\xi'' \in \mathsf{Set}(\xi)} W(\xi'', \nu) \right| \leq \frac{r}{1-\zeta}, \tag{23}$$

which exactly implies $|\mathcal{W}(\mu, \nu) - \mathcal{W}(\xi, \nu)| \leq \frac{r}{1-\zeta}$.

According to [Ding et al., 2021], the output coreset $S$ is an $r$-cover for $Q$ under the Wasserstein distance. This means that for any $\mu \in Q$, there exists $\xi \in S$ such that $W(\mu, \xi) \leq r$. Consequently, for any $\mu \in Q$, there exists $\xi \in S$ such that $|\mathcal{W}(\mu, \nu) - \mathcal{W}(\xi, \nu)| \leq \frac{r}{1-\zeta}$.

Let $g(\cdot) = \mathcal{W}(\mu, \cdot)$ and $f(\cdot) = \mathcal{W}(\xi, \cdot)$. Using Lemma C.5, we obtain

$$\left| \min_{\nu \in Q} \mathcal{W}(\mu, \nu) - \min_{\nu' \in Q} \mathcal{W}(\xi, \nu') \right| \leq \frac{r}{1-\zeta}, \tag{24}$$

which implies $|\mathcal{W}(\mu, C) - \mathcal{W}(\xi, C)| \leq \frac{r}{1-\zeta}$.

By setting $g(\cdot) = -\mathcal{W}(\cdot, C)$, we have $|\max_{\mu \in Q} \mathcal{W}(\mu, C) - \max_{\xi \in S} \mathcal{W}(\xi, C)| \leq \frac{r}{1-\zeta}$ according to Lemma C.4.

Setting $r = \mathcal{O}(\epsilon\Gamma)$, we obtain

$$|\mathsf{cost}(Q, C) - \mathsf{cost}(S, C)| \leq \epsilon \cdot \mathsf{cost}(Q, C). \tag{25}$$

**Time complexity:** Algorithm 2 induces a tree with a maximum height of $\mathcal{O}(\log \frac{R}{r})$. Constructing each layer requires time $\mathcal{O}(2^{2 \cdot \mathsf{ddim}} \cdot |Q| \cdot \mathcal{T})$, thus the total time complexity is $\mathcal{O}(2^{2 \cdot \mathsf{ddim}} \cdot |Q| \cdot \mathcal{T} \cdot \log \frac{R}{r})$.

**Coreset size:** After executing the Gonzalez algorithm $\mathcal{O}(2^{2 \cdot \mathsf{ddim}})$ rounds, the radius is reduced by half. This implies that the degree of the tree is at most $\mathcal{O}(2^{2 \cdot \mathsf{ddim}})$. Therefore, the coreset size, which corresponds to the total number of nodes in the tree, is $\mathcal{O}((\frac{R}{r})^{2 \cdot \mathsf{ddim}})$.

$\square$

By setting $g(\cdot) = \mathsf{cost}(Q, \cdot)$ and $f(\cdot) = \mathsf{cost}(S, \cdot)$, we obtain the following corollary according to Lemma C.5.

**Corollary 4.3.** *Algorithm 2 takes $Q$ as its input and outputs the coreset $S$. The output $S$ satisfies the following property*

$$\left| \min_{C \in \mathcal{P}(\mathcal{X}), |C|=k} \mathsf{cost}(Q, C) - \min_{C' \in \mathcal{P}(\mathcal{X}), |C'|=k} \mathsf{cost}(S, C') \right| \leq \min_{C \in \mathcal{P}(\mathcal{X}), |C|=k} \epsilon \cdot \mathsf{cost}(Q, C).$$

## D  Full Experiments

This section demonstrates the effectiveness of our RWC-clustering problem, the efficiency of the coreset method, the necessity of the seeding algorithm, and several ablation studies.

All the experiments were performed on a server with an AMD EPYC 9754 128-Core Processor with 18 vCPUs, 60GB of RAM, and Python 3.12. The server utilized an RTX 4090D GPU with 24GB of VRAM. We used the POT library [Flamary et al., 2021] to compute the Wasserstein distance (WD) [Bonneel et al., 2011] and unbalanced optimal transport (UOT) [Chizat et al., 2018, Frogner et al., 2015]. To improve numerical stability, we adopted the stabilized Sinkhorn algorithm [Schmitzer, 2019, Chizat et al., 2017] for computing the entropy regularization version. The reported results are averaged over five runs.

We validated our methods on the following datasets.

i) **Geometric shapes** is a toy dataset designed by us, consisting of five geometric shapes. It is used to verify the advantages of our seeding algorithm and the RWC-clustering problem intuitively. Each shape is represented by a point set.

ii) **MNIST** [LeCun et al., 1998] is a well-known handwritten digit dataset. For each image, we extract the pixels with higher grayscale values (greater than 0.3) to form the corresponding point set.

iii) **ModelNet10** [Wu et al., 2015] is a standard dataset containing 3D CAD models. Each CAD model is discretized and represented as a point set.

iv) **KITTI** dataset [Geiger et al., 2012] is a widely used benchmark for autonomous driving, containing 3D LiDAR scans. We use its 3D object detection subset, which contains point clouds for various categories such as *Pedestrian*, *Cyclist*, *Car*, *Van*, *Truck*, *Person_sitting*, *Tram*, and *Misc*.

v) **ShapeNetCore** [Chang et al., 2015] is a dataset containing a large collection of 3D object models. It includes 55 categories, such as chairs, tables, cars, airplanes, and other common objects.

vi) **ScanObjectNN** [Uy et al., 2019] is a real-world 3D object classification benchmark, which is captured from real scans, making it more challenging than synthetic datasets such as ModelNet10.

vii) **nuScenes Mini** [Caesar et al., 2020] is a subset of the full nuScenes dataset, containing 10 scenes and providing high-quality 3D point cloud data with corresponding annotations.

We extract object-level point cloud instances from the dataset and uniformly sample 300 points from each to construct our point cloud dataset. If the original point count is smaller than $n$, we perform uniform sampling with replacement; otherwise, we apply uniform sampling without replacement.

For the point set $\{x_i\}_{i \in [n]}$ corresponding to a specific data item in the above datasets, we represent it as a probability measure $\mu = \sum_{i=1}^{n} \frac{1}{n} \delta_{x_i}$ to construct the clean dataset $Q^0$. The noisy dataset $Q$ is generated by adding clustered noise with $\zeta$ mass following a Gaussian distribution $\mathcal{N}(\mu, \sigma^2)$ to each clean probability measure.

To evaluate the performance of our methods, we consider the following three criteria: i) Runtime: This includes the sampling time and the clustering time required to compute the $k$-center set $C$. ii) cost-cd: Defined as $\max_{\mu \in Q^0} \min_{\nu \in C} W(\mu, \nu)$, it quantifies the distance between the center set $C$ and the original clean dataset $Q^0$. iii) cost-nd: Defined as $\max_{\mu \in Q} \min_{\nu \in C} \mathcal{W}(\mu, \nu)$, it evaluates the distance from center set $C$ to the noisy dataset $Q$. The baselines for these three criteria are established using the results computed on the original full dataset for comparison.

## D.1 Effectiveness of our RWC-clustering problem

To demonstrate the effectiveness of the RWC-clustering problem, we conducted a series of experiments on several datasets. We compare the clustering quality computed by UOT-based clustering, WD-based clustering, and RWC-clustering. All three methods follow the same framework, where seeding is used for initialization, followed by a local search refinement. Specifically, the UOT-based clustering algorithm is derived by replacing RWD with UOT in Algorithms 1 and 4. Since WD is a metric, thus the WD-based clustering can directly use the existing Gonzalez's algorithm [Gonzalez, 1985] along with the local search method [Lattanzi and Sohler, 2019, Choo et al., 2020].

We first present results on several real-world datasets. Then, to provide a more intuitive understanding of our method, we illustrate the results on a toy dataset.

**Effectiveness of our RWC-clustering problem on real-world datasets:**

We randomly select 200 data items from each dataset to serve as the experimental data. Table 2 evaluates the effectiveness of our RWC-clustering problem across different real-world datasets. Among these methods, our RWC-clustering approach achieves the best denoising performance, outperforming the other two methods in almost all datasets.

Due to the randomness inherent in our algorithms and the approximate nature of the derived $k$-center set, some fluctuations are inevitable. A slightly inferior performance on the cost-cd metric for MNIST is acceptable.

Table 2: Comparison of our RWC-clustering with UOT-based clustering and WD-based clustering on different datasets. We fix the dataset size as 200, $k = 10$, $Z = 200$, $\sigma = 1$, $\tau = 10$ and $\zeta = 0.1$, where $Z$ is the number of iterations in post-processing procedure (i.e., Algorithm 4).

| Dataset | Method | cost-nd($\downarrow$) | cost-cd($\downarrow$) | Runtime ($\downarrow$) |
|---|---|---|---|---|
| ShapeNetCore | RWC-clustering | $\mathbf{5.42}_{\pm 0.15}$ | $\mathbf{5.97}_{\pm 0.42}$ | $485.78_{\pm 0.43}$ |
| | UOT-based clustering | $13.25_{\pm 0.31}$ | $15.91_{\pm 1.83}$ | $447.76_{\pm 9.74}$ |
| | WD-based clustering | $8.69_{\pm 0.89}$ | $10.92_{\pm 0.94}$ | $485.78_{\pm 0.43}$ |
| ScanObjectNN | RWC-clustering | $\mathbf{3.67}_{\pm 0.37}$ | $\mathbf{5.78}_{\pm 0.00}$ | $519.01_{\pm 1.62}$ |
| | UOT-based clustering | $8.55_{\pm 0.15}$ | $11.97_{\pm 0.00}$ | $447.66_{\pm 5.44}$ |
| | WD-based clustering | $6.96_{\pm 0.78}$ | $13.10_{\pm 0.00}$ | $519.01_{\pm 1.62}$ |
| nuScenes Mini | RWC-clustering | $\mathbf{4.96}_{\pm 0.20}$ | $\mathbf{5.18}_{\pm 0.12}$ | $529.29_{\pm 4.27}$ |
| | UOT-based clustering | $9.00_{\pm 2.98}$ | $9.89_{\pm 3.46}$ | $465.98_{\pm 26.43}$ |
| | WD-based clustering | $13.20_{\pm 0.23}$ | $17.54_{\pm 0.60}$ | $529.29_{\pm 4.27}$ |
| ModelNet10 | RWC-clustering | $\mathbf{2.20}_{\pm 0.13}$ | $\mathbf{2.40}_{\pm 0.05}$ | $475.98_{\pm 2.39}$ |
| | UOT-based clustering | $4.07_{\pm 0.30}$ | $4.26_{\pm 0.40}$ | $446.11_{\pm 7.66}$ |
| | WD-based clustering | $5.32_{\pm 0.48}$ | $7.25_{\pm 0.72}$ | $475.98_{\pm 2.39}$ |
| MNIST | RWC-clustering | $\mathbf{6.80}_{\pm 0.06}$ | $12.37_{\pm 0.45}$ | $420.84_{\pm 0.66}$ |
| | UOT-based clustering | $9.69_{\pm 0.62}$ | $\mathbf{11.14}_{\pm 1.30}$ | $443.34_{\pm 9.42}$ |
| | WD-based clustering | $9.37_{\pm 0.13}$ | $17.02_{\pm 0.19}$ | $420.84_{\pm 0.66}$ |
| KITTI | RWC-clustering | $\mathbf{3.36}_{\pm 0.13}$ | $\mathbf{4.14}_{\pm 0.00}$ | $532.82_{\pm 0.49}$ |
| | UOT-based clustering | $10.84_{\pm 0.99}$ | $11.86_{\pm 0.40}$ | $447.57_{\pm 11.84}$ |
| | WD-based clustering | $7.83_{\pm 0.21}$ | $11.51_{\pm 0.68}$ | $532.82_{\pm 0.49}$ |

Table 3 shows that among these methods, our RWC-clustering approach consistently achieves the best denoising performance, outperforming the other two methods across different values of the parameter $k$.

Table 3: Comparison of our RWC-clustering with UOT-based clustering and WD-based clustering on ModelNet10 dataset across different $k$. We fix the dataset size as 200, $Z = 200, \sigma = 1, \tau = 10$ and $\zeta = 0.1$, where $Z$ is the number of iterations in post-processing procedure (i.e., Algorithm 4).

| $k$ | Method | cost-nd($\downarrow$) | cost-cd($\downarrow$) | Runtime ($\downarrow$) |
|---|---|---|---|---|
| 10 | RWC-clustering | $\mathbf{2.20}_{\pm 0.13}$ | $\mathbf{2.40}_{\pm 0.05}$ | $475.98_{\pm 2.39}$ |
| | UOT-based clustering | $4.07_{\pm 0.30}$ | $4.26_{\pm 0.40}$ | $446.11_{\pm 7.66}$ |
| | WD-based clustering | $5.32_{\pm 0.48}$ | $7.25_{\pm 0.72}$ | $475.98_{\pm 2.39}$ |
| 20 | RWC-clustering | $\mathbf{1.84}_{\pm 0.08}$ | $\mathbf{2.11}_{\pm 0.06}$ | $526.83_{\pm 1.67}$ |
| | UOT-based clustering | $4.01_{\pm 0.28}$ | $4.37_{\pm 0.24}$ | $573.45_{\pm 1.52}$ |
| | WD-based clustering | $5.34_{\pm 0.55}$ | $6.35_{\pm 0.65}$ | $526.83_{\pm 1.67}$ |
| 30 | RWC-clustering | $\mathbf{1.52}_{\pm 0.05}$ | $\mathbf{1.73}_{\pm 0.10}$ | $575.72_{\pm 3.28}$ |
| | UOT-based clustering | $3.85_{\pm 0.24}$ | $4.08_{\pm 0.02}$ | $511.92_{\pm 2.50}$ |
| | WD-based clustering | $4.51_{\pm 0.27}$ | $6.08_{\pm 0.12}$ | $575.72_{\pm 5.28}$ |

Table 3 shows that among these methods, our RWC-clustering approach consistently achieves the best denoising performance, outperforming the other two methods across different values of the parameter $\zeta$.

Table 4: Comparison of our RWC-clustering with UOT-based clustering and WD-based clustering on ModelNet10 dataset across different mass of noise. We fix the dataset size as 200, $k = 10$, $Z = 200, \sigma = 1$ and $\tau = 10$, where $Z$ is the number of iterations in post-processing procedure (i.e., Algorithm 4).

| $\zeta$ | Method | cost-nd($\downarrow$) | cost-cd($\downarrow$) | Runtime ($\downarrow$) |
|---|---|---|---|---|
| 0.1 | RWC-clustering | $\mathbf{2.20}_{\pm 0.13}$ | $\mathbf{2.40}_{\pm 0.05}$ | $475.98_{\pm 2.39}$ |
| | UOT-based clustering | $4.07_{\pm 0.30}$ | $4.26_{\pm 0.40}$ | $446.11_{\pm 7.66}$ |
| | WD-based clustering | $5.32_{\pm 0.48}$ | $7.25_{\pm 0.72}$ | $475.98_{\pm 2.39}$ |
| 0.2 | RWC-clustering | $\mathbf{2.23}_{\pm 0.12}$ | $\mathbf{2.76}_{\pm 0.19}$ | $650.75_{\pm 7.80}$ |
| | UOT-based clustering | $3.94_{\pm 0.17}$ | $4.28_{\pm 0.27}$ | $431.84_{\pm 11.77}$ |
| | WD-based clustering | $8.83_{\pm 1.37}$ | $11.77_{\pm 1.23}$ | $650.75_{\pm 7.80}$ |
| 0.3 | RWC-clustering | $\mathbf{2.21}_{\pm 0.11}$ | $\mathbf{3.24}_{\pm 1.02}$ | $841.29_{\pm 0.46}$ |
| | UOT-based clustering | $3.72_{\pm 0.12}$ | $4.34_{\pm 0.15}$ | $432.44_{\pm 10.84}$ |
| | WD-based clustering | $11.71_{\pm 1.89}$ | $16.37_{\pm 1.75}$ | $841.29_{\pm 0.46}$ |

**Effectiveness of our RWC-clustering problem on toy dataset:**

We visualized the clean geometric shapes in Figure 11(a), and the noisy geometric shapes are in Figure 3.

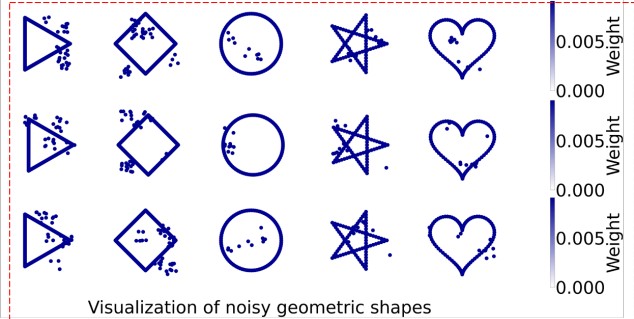

Figure 3: Visualization of noisy geometric shapes.

Figure 4 evaluates the **effectiveness of our RWC-clustering problem**. Specifically, Figure 4(a) displays the clustering results obtained using our approach, while Figure 4(b) shows results based on Unbalanced Optimal Transport (UOT), and Figure 4(c) presents clustering results using the classical Wasserstein distance (WD).

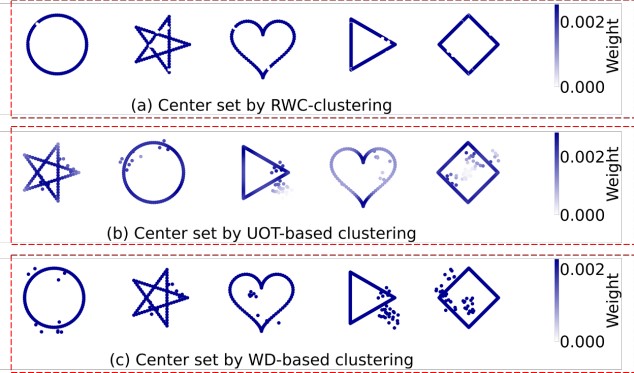

Figure 4: Comparing our RWC-clustering with WD-based clustering and UOT-based clustering.

Among these, the WD-based clustering exhibits the worst performance, showing almost no denoising capability. The UOT-based clustering outperforms the WD-based clustering; however, the inclusion of an entropy regularization term causes a diffusion effect that hinders noise removal, leaving some residual noise inescapably. In contrast, our RWC-clustering approach achieves the best denoising performance, outperforming the other two methods.

## D.2 Effectiveness of our coreset method

We show the effectiveness of our coreset (CS) method by comparing it against the uniform sampling (US) method. Specifically, we first construct a coreset $S \subseteq Q$ and simultaneously generate a uniformly sampled subset $S'$ of the same size $|S|$ from the full dataset $Q$ as a baseline. We then run the clustering algorithm ( i.e., Algorithms 1 and 4 ) on both subsets. If the clustering algorithm is applied to the coreset $S$, we refer to it as the CS method. Conversely, if it is applied to the uniformly sampled subset $S'$, we refer to it as the US method. Since the exact size of the coreset cannot be pre-specified, to ensure fairness between the two sampling methods (SM), we set the sample size (SS) of the US method to match that of the CS method. The green dashed line indicates the results on the full dataset, which serves as the baseline for comparison.

**Coreset method on MNIST dataset:**

Figure 5 illustrates the performance of our CS method on the MNIST dataset across different sample size. Although our CS method is slightly more time-consuming compared to the US method, it remains significantly more efficient than processing the original dataset. Moreover, in terms of both cost-cd and cost-nd criteria, our CS method consistently outperforms the US method.

Due to the randomness inherent in our algorithms and the approximate nature of the derived $k$-center set, some fluctuations are inevitable.

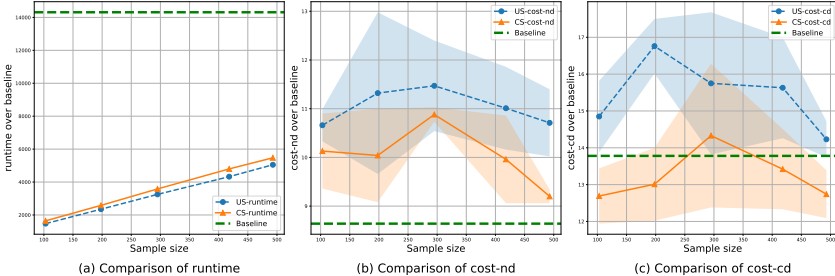

(a) Comparison of runtime    (b) Comparison of cost-nd    (c) Comparison of cost-cd

Figure 5: Comparison of the US method and our CS method across varying sample sizes on MNIST dataset. We fix $k = 10, Z = 200, \sigma = 1, \tau = 10$ and $\zeta = 0.1$, where $Z$ is the number of iterations in post-processing procedure (i.e., Algorithm 4).

Results on different $k$: From Table 5, our CS method consistently has an advantage over the US method across different $k$.

When using the full dataset, the large data volume makes it difficult for the 200-round local search to adequately explore the entire dataset. As a result, in some cases, the performance on the full dataset is actually worse than that on our small-size coreset. Our coreset method places more attention on boundary points, enabling it to better capture the diversity of the dataset. Consequently, it not only achieves higher computational efficiency but also achieves better clustering quality.

Table 5: Comparison of the US method and our CS method with varying values of $k$ on MNIST dataset. We fix the sample size as 198, $Z = 200, \sigma = 1, \tau = 10$ and $\zeta = 0.1$, where $Z$ is the number of iterations in post-processing procedure (i.e., Algorithm 4).

| $k$ | SM | cost-nd($\downarrow$) | cost-cd($\downarrow$) | Runtime($\downarrow$) |
|---|---|---|---|---|
| 10 | US | $11.32_{\pm 1.66}$ | $16.76_{\pm 0.74}$ | $2350.03_{\pm 33.77}$ |
| | CS | $\mathbf{10.04}_{\pm 0.96}$ | $\mathbf{13.01}_{\pm 0.99}$ | $2582.41_{\pm 40.61}$ |
| 20 | US | $9.85_{\pm 0.72}$ | $14.75_{\pm 1.07}$ | $2443.48_{\pm 59.73}$ |
| | CS | $\mathbf{8.54}_{\pm 0.66}$ | $\mathbf{10.81}_{\pm 0.22}$ | $2741.75_{\pm 49.83}$ |
| 30 | US | $9.81_{\pm 0.30}$ | $15.30_{\pm 0.77}$ | $2584.93_{\pm 72.05}$ |
| | CS | $\mathbf{8.35}_{\pm 0.34}$ | $\mathbf{11.24}_{\pm 0.71}$ | $2882.62_{\pm 67.72}$ |

**Coreset method on ModelNet10 dataset:**

Figure 6 illustrates the performance of our CS method on the ModelNet10 dataset across different sample size. Our CS method is slightly more time-consuming than the US method. However, it remains significantly more efficient than processing the original dataset. Moreover, in terms of both cost-cd and cost-nd criteria, our CS method consistently outperforms the US method.

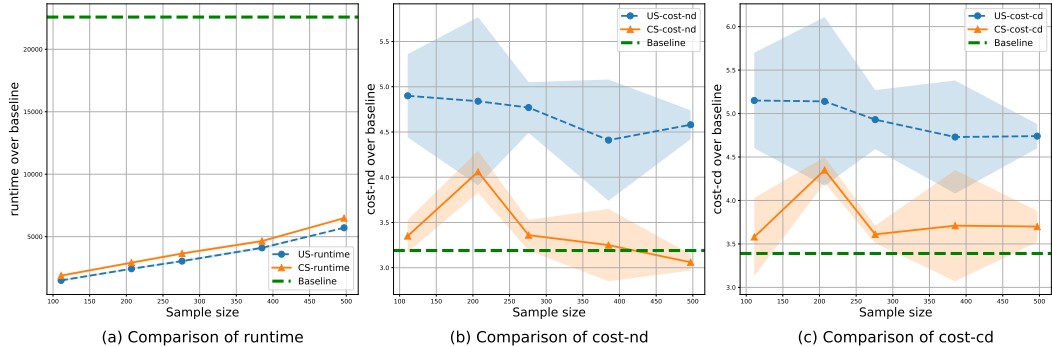

(a) Comparison of runtime     (b) Comparison of cost-nd     (c) Comparison of cost-cd

Figure 6: Comparison of the US method and our CS method across varying sample sizes on ModelNet10 dataset. We fix $k = 10, Z = 200, \sigma = 1, \tau = 10$ and $\zeta = 0.1$, where $Z$ is the number of iterations in post-processing procedure (i.e., Algorithm 4).

Results on different $k$: From Table 6, our CS method consistently has an advantage over the US method across different $k$.

Table 6: Comparison of the US method and our CS method with varying values of $k$ on ModelNet10 dataset. We fix the sample size as 276, $Z = 200, \sigma = 1, \tau = 10$ and $\zeta = 0.1$, where $Z$ is the number of iterations in post-processing procedure (i.e., Algorithm 4).

| $k$ | SM | cost-nd($\downarrow$) | cost-cd($\downarrow$) | Runtime($\downarrow$) |
|---|---|---|---|---|
| 10 | US | $4.77_{\pm 0.28}$ | $4.93_{\pm 0.34}$ | $3044.64_{\pm 84.37}$ |
| | CS | $\mathbf{3.36}_{\pm 0.17}$ | $\mathbf{3.61}_{\pm 0.10}$ | $3563.36_{\pm 97.92}$ |
| 20 | US | $3.78_{\pm 0.24}$ | $4.13_{\pm 0.40}$ | $3306.95_{\pm 38.13}$ |
| | CS | $\mathbf{2.74}_{\pm 0.10}$ | $\mathbf{3.47}_{\pm 0.06}$ | $3800.35_{\pm 41.62}$ |
| 30 | US | $3.92_{\pm 0.02}$ | $4.36_{\pm 0.01}$ | $3531.90_{\pm 51.19}$ |
| | CS | $\mathbf{2.45}_{\pm 0.08}$ | $\mathbf{2.90}_{\pm 0.51}$ | $4009.34_{\pm 40.75}$ |

**Coreset method on KITTI dataset:**

Figure 7 illustrates the performance of our CS method on the KITTI dataset across different sample size. Our CS method consistently outperforms the US method on both cost-cd and cost-nd criteria.

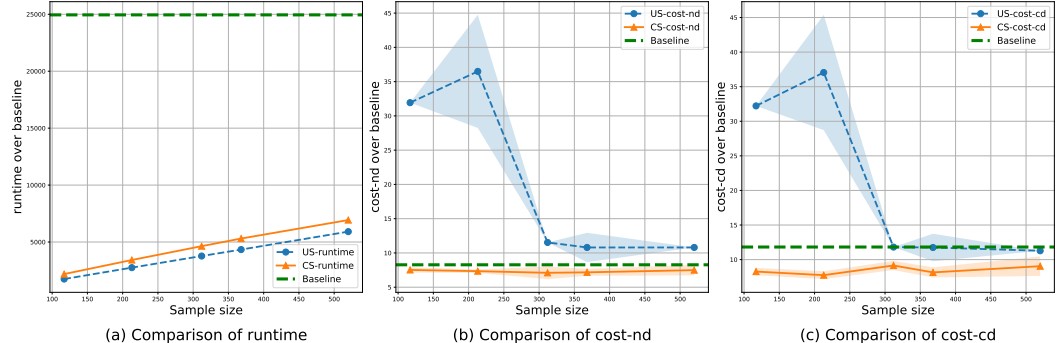

(a) Comparison of runtime  (b) Comparison of cost-nd  (c) Comparison of cost-cd

Figure 7: Comparison of the US method and our CS method across varying sample sizes on KITTI dataset. We fix $k = 10, Z = 200, \sigma = 1, \tau = 10$ and $\zeta = 0.1$, where $Z$ is the number of iterations in post-processing procedure (i.e., Algorithm 4).

 Results on different $k$: From Table 7, our CS method consistently has an advantage over the US method across different $k$.

Table 7: Comparison of the US method and our CS method with varying values of $k$ on KITTI dataset. We fix the sample size as 213, $Z = 200, \sigma = 1, \tau = 10$ and $\zeta = 0.1$, where $Z$ is the number of iterations in post-processing procedure (i.e., Algorithm 4).

| $k$ | SM | cost-nd($\downarrow$) | cost-cd($\downarrow$) | Runtime($\downarrow$) |
|---|---|---|---|---|
| 10 | US | $36.49_{\pm 8.25}$ | $37.05_{\pm 8.33}$ | $2750.14_{\pm 7.69}$ |
|    | CS | $\mathbf{7.34}_{\pm 0.25}$ | $\mathbf{7.75}_{\pm 0.57}$ | $3419.77_{\pm 2.97}$ |
| 20 | US | $29.38_{\pm 1.96}$ | $29.82_{\pm 1.92}$ | $2907.22_{\pm 7.01}$ |
|    | CS | $\mathbf{4.34}_{\pm 0.93}$ | $5.60_{\pm 0.87}$ | $3601.32_{\pm 2.22}$ |
| 30 | US | $12.56_{\pm 0.00}$ | $13.34_{\pm 0.00}$ | $2989.84_{\pm 15.69}$ |
|    | CS | $\mathbf{2.84}_{\pm 0.33}$ | $\mathbf{3.60}_{\pm 0.29}$ | $3745.60_{\pm 50.50}$ |

**Coreset method on ShapeNetCore dataset:**

Figure 8 illustrates the performance of our CS method on the ShapeNetCore dataset across different sample size. Our CS method consistently outperforms the US method on both cost-cd and cost-nd criteria.

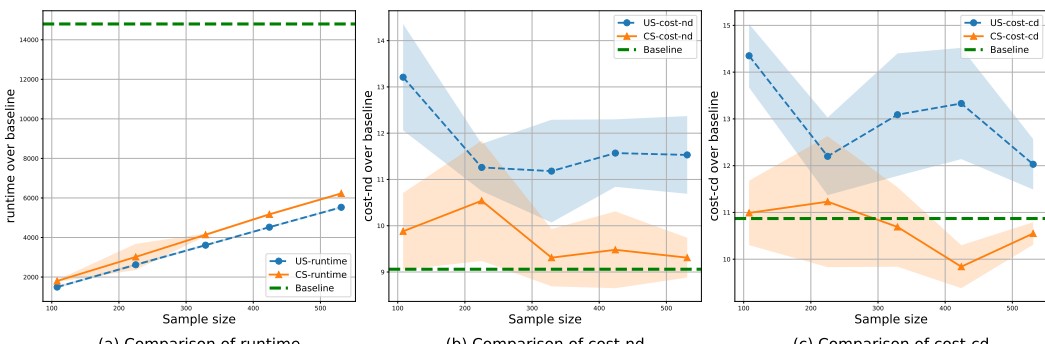

(a) Comparison of runtime     (b) Comparison of cost-nd     (c) Comparison of cost-cd

Figure 8: Comparison of the US method and our CS method across varying sample sizes on ShapeNetCore dataset. We fix $k = 10, Z = 200, \sigma = 1, \tau = 10$ and $\zeta = 0.1$, where $Z$ is the number of iterations in post-processing procedure (i.e., Algorithm 4).

Results on different $k$: From Table 8, our CS method consistently has an advantage over the US method across different $k$.

Table 8: Comparison of the US method and our CS method with varying values of $k$ on ShapeNetCore dataset. We fix the sample size as 225, $Z = 200, \sigma = 1, \tau = 10$ and $\zeta = 0.1$, where $Z$ is the number of iterations in post-processing procedure (i.e., Algorithm 4).

| $k$ | SM | cost-nd($\downarrow$) | cost-cd($\downarrow$) | Runtime($\downarrow$) |
|---|---|---|---|---|
| 10 | US | $11.26_{\pm 0.51}$ | $12.20_{\pm 0.83}$ | $2621.13_{\pm 66.35}$ |
|    | CS | $\mathbf{10.54}_{\pm 1.30}$ | $\mathbf{11.23}_{\pm 1.40}$ | $2900.54_{\pm 662.23}$ |
| 20 | US | $8.98_{\pm 0.33}$ | $9.89_{\pm 0.25}$ | $2823.81_{\pm 6.63}$ |
|    | CS | $\mathbf{8.63}_{\pm 0.79}$ | $\mathbf{9.74}_{\pm 0.53}$ | $3320.58_{\pm 19.00}$ |
| 30 | US | $9.23_{\pm 0.76}$ | $10.31_{\pm 0.87}$ | $3022.49_{\pm 30.35}$ |
|    | CS | $\mathbf{7.63}_{\pm 0.07}$ | $\mathbf{8.54}_{\pm 0.21}$ | $3486.17_{\pm 29.55}$ |

**Coreset method on ScanObjectnn dataset:**

Figure 9 illustrates the performance of our CS method on the ScanObjectnn dataset across different sample size. Our CS method consistently outperforms the US method on both cost-cd and cost-nd criteria.

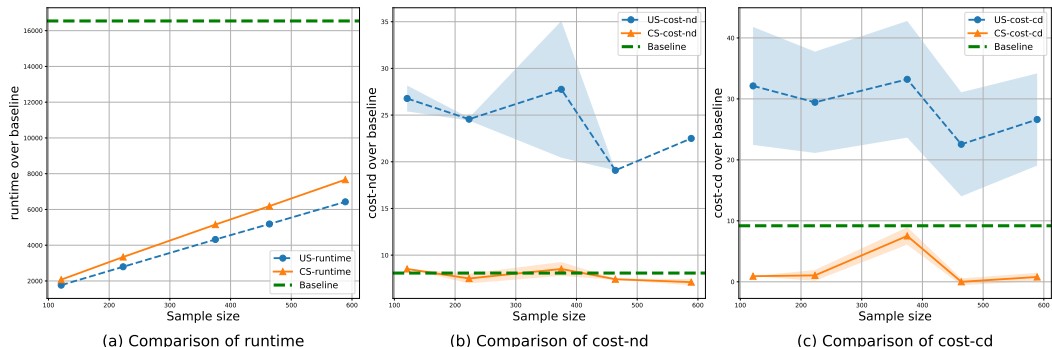

(a) Comparison of runtime       (b) Comparison of cost-nd       (c) Comparison of cost-cd

Figure 9: Comparison of the US method and our CS method across varying sample sizes on ScanObjectnn dataset. We fix $k = 10, Z = 200, \sigma = 1, \tau = 10$ and $\zeta = 0.1$, where $Z$ is the number of iterations in post-processing procedure (i.e., Algorithm 4).

Results on different $k$: From Table 9, our CS method consistently has an advantage over the US method across different $k$.

Table 9: Comparison of the US method and our CS method with varying values of $k$ on ScanObjectnn dataset. We fix the sample size as 223, $Z = 200, \sigma = 1, \tau = 10$ and $\zeta = 0.1$, where $Z$ is the number of iterations in post-processing procedure (i.e., Algorithm 4).

| $k$ | SM | cost-nd($\downarrow$) | cost-cd($\downarrow$) | Runtime($\downarrow$) |
|---|---|---|---|---|
| 10 | US | $24.56_{\pm 0.15}$ | $29.44_{\pm 1.05}$ | $2793.50_{\pm 0.49}$ |
|    | CS | $\mathbf{7.50}_{\pm 0.53}$ | $\mathbf{8.31}_{\pm 0.89}$ | $3340.34_{\pm 6.32}$ |
| 20 | US | $19.20_{\pm 0.00}$ | $22.98_{\pm 0.00}$ | $2886.22_{\pm 24.60}$ |
|    | CS | $\mathbf{5.27}_{\pm 0.45}$ | $\mathbf{5.98}_{\pm 0.14}$ | $3542.27_{\pm 14.56}$ |
| 30 | US | $19.20_{\pm 0.00}$ | $22.98_{\pm 0.00}$ | $3077.61_{\pm 13.00}$ |
|    | CS | $\mathbf{4.01}_{\pm 0.22}$ | $\mathbf{4.81}_{\pm 0.33}$ | $3623.86_{\pm 4.22}$ |

**Coreset method on nuScenes Mini dataset:**

Figure 10 illustrates the performance of our CS method on the nuScenes Mini dataset across different sample size. Our CS method consistently outperforms the US method on both cost-cd and cost-nd criteria.

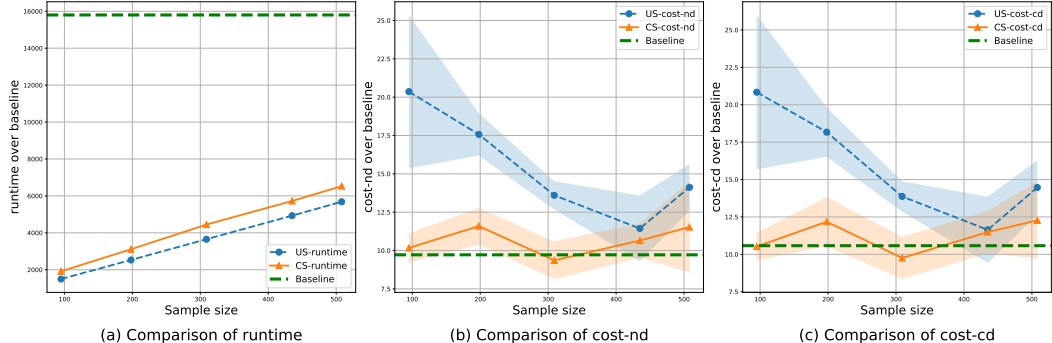

(a) Comparison of runtime     (b) Comparison of cost-nd     (c) Comparison of cost-cd

Figure 10: Comparison of the US method and our CS method across varying sample sizes on nuScenes Mini dataset. We fix $k = 10, Z = 200, \sigma = 1, \tau = 10$ and $\zeta = 0.1$, where $Z$ is the number of iterations in post-processing procedure (i.e., Algorithm 4).

Results on different $k$: From Table 10, our CS method consistently has an advantage over the US method across different $k$.

Table 10: Comparison of the US method and our CS method with varying values of $k$ on nuScenes Mini dataset. We fix the sample size as 198, $Z = 200, \sigma = 1, \tau = 10$ and $\zeta = 0.1$, where $Z$ is the number of iterations in post-processing procedure (i.e., Algorithm 4 in appendix)

| $k$ | SM | cost-nd($\downarrow$) | cost-cd($\downarrow$) | Runtime($\downarrow$) |
|---|---|---|---|---|
| 10 | US | $17.57_{\pm 1.38}$ | $18.17_{\pm 1.65}$ | $2530.11_{\pm 11.61}$ |
|  | CS | $\mathbf{11.59}_{\pm 1.20}$ | $\mathbf{12.18}_{\pm 1.64}$ | $3066.76_{\pm 10.40}$ |
| 20 | US | $13.04_{\pm 0.05}$ | $13.29_{\pm 0.00}$ | $2662.22_{\pm 37.98}$ |
|  | CS | $\mathbf{6.03}_{\pm 0.31}$ | $\mathbf{6.20}_{\pm 0.39}$ | $3199.50_{\pm 34.17}$ |
| 30 | US | $13.02_{\pm 0.00}$ | $13.44_{\pm 0.25}$ | $2821.26_{\pm 46.32}$ |
|  | CS | $\mathbf{5.73}_{\pm 0.92}$ | $\mathbf{5.85}_{\pm 1.01}$ | $3350.30_{\pm 34.08}$ |

### D.3 Ablation experiments

**Ablation experiments on $\tau$:**

We conduct ablation experiments on the ModelNet10 dataset to explore the effect of the hyperparameter $\tau$. In this experiment, we randomly select 200 samples from ModelNet10 to form our dataset. As shown in Table 11, setting $\tau$ too small or too large leads to suboptimal performance. A too small $\tau$ may result in degraded clustering quality, while a large $\tau$ significantly increases computational cost without consistent improvements in quality. Therefore, we recommend selecting a relatively small constant for $\tau$. In all our experiments, we fix $\tau = 10$ for providing a good balance between efficiency and performance.

Table 11: Comparison of our algorithm (i.e., Algorithm 1 and 4) for solving RWC-clustering problem by using varying parameter $\tau$ on ModelNet10 dataset. We fix the dataset size as 200, $k = 10, Z = 200, \sigma = 1$ and $\zeta = 0.1$, where $Z$ is the number of iterations in post-processing procedure (i.e., Algorithm 4).

| $\tau$ | cost-nd($\downarrow$) | cost-cd($\downarrow$) | Runtime ($\downarrow$) |
|---|---|---|---|
| 1 | $2.65_{\pm 0.40}$ | $4.19_{\pm 0.70}$ | $1419.53_{\pm 352.59}$ |
| 2 | $2.26_{\pm 0.04}$ | $2.57_{\pm 0.05}$ | $1524.12_{\pm 436.52}$ |
| 3 | $2.20_{\pm 0.04}$ | $2.48_{\pm 0.05}$ | $1701.30_{\pm 476.68}$ |
| 5 | $2.35_{\pm 0.17}$ | $2.57_{\pm 0.16}$ | $2025.11_{\pm 179.11}$ |
| 10 | $\mathbf{2.18}_{\pm 0.04}$ | $\mathbf{2.43}_{\pm 0.10}$ | $2498.92_{\pm 125.57}$ |
| 20 | $2.26_{\pm 0.10}$ | $2.48_{\pm 0.14}$ | $3874.76_{\pm 91.72}$ |
| 50 | $2.32_{\pm 0.17}$ | $2.52_{\pm 0.07}$ | $11997.44_{\pm 1142.86}$ |

**Selection of $\zeta$ under unknown noise mass:**

This experiment aims to explore how to select the parameter $\zeta$ when the true noise mass is unknown. We randomly sample 200 samples from the ModelNet10 to form the dataset, each data item containing 0.3 mass of noise. However, we run our algorithm across different $\zeta$.

Table 12: Comparison of our algorithm (i.e., Algorithm 1 and 4) for solving RWC-clustering problem across different values of $\zeta$ on ModelNet10 dataset with 0.3 mass of noise. We fix the dataset size as 200, $k = 10, Z = 200, \sigma = 1$ and $\tau = 10$, where $Z$ is the number of iterations in post-processing procedure (i.e., Algorithm 4).

| $\zeta$ | cost-nd($\downarrow$) | cost-cd($\downarrow$) | Runtime($\downarrow$) |
|---|---|---|---|
| 0.05 | $13.48_{\pm 0.15}$ | $13.58_{\pm 1.48}$ | $2387.10_{\pm 8.16}$ |
| 0.1 | $9.99_{\pm 0.11}$ | $10.90_{\pm 1.51}$ | $2861.19_{\pm 189.57}$ |
| 0.2 | $5.10_{\pm 0.02}$ | $5.45_{\pm 0.69}$ | $2985.63_{\pm 226.70}$ |
| 0.3 | $\textcolor{red}{2.29}_{\pm 0.17}$ | $\textcolor{red}{3.58}_{\pm 0.94}$ | $2998.34_{\pm 166.96}$ |
| 0.4 | $1.72_{\pm 0.07}$ | $3.26_{\pm 0.36}$ | $2943.35_{\pm 35.42}$ |
| 0.5 | $1.43_{\pm 0.08}$ | $3.86_{\pm 0.49}$ | $3055.85_{\pm 88.55}$ |
| 0.6 | $1.21_{\pm 0.06}$ | $5.01_{\pm 0.59}$ | $3122.77_{\pm 112.56}$ |
| 0.7 | $1.00_{\pm 0.04}$ | $4.46_{\pm 0.37}$ | $3589.76_{\pm 26.52}$ |
| 0.8 | $0.75_{\pm 0.01}$ | $4.93_{\pm 0.08}$ | $3722.17_{\pm 16.04}$ |
| 0.9 | $0.59_{\pm 0.02}$ | $7.30_{\pm 1.00}$ | $3921.55_{\pm 14.58}$ |

Nietert et al. [2022] theoretically demonstrates that when $\zeta \in [0, 1)$ is slightly overestimated relative to the true noise mass, the optimal solution can still be attained. This suggests that a mild overestimation of $\zeta$ does not significantly affect the results.

Our results in Table 13 confirm that slightly overestimating $\zeta$ has only a minor impact, while underestimating it severely degrades the solution quality. Therefore, when the true noise mass is unknown, we recommend setting $\zeta$ slightly larger than the expected noise mass to ensure robust performance.

**Ablation experiment for our CS method across varying mass $\zeta$ of noise:**

Table 13 illustrates that the CS method consistently outperforms the US method under both cost-cd and cost-nd criteria across varying mass $\zeta$ of noise.

Table 13: Comparison of the US method and our CS method using varying mass $\zeta$ of noise. We fix $k = 10, Z = 200, \sigma = 1$ and $\tau = 10$, where $Z$ is the number of iterations in post-processing procedure (i.e., Algorithm 4 in appendix)

| $\zeta$ | SM | SS | cost-nd($\downarrow$) | cost-cd($\downarrow$) | Runtime ($\downarrow$) |
|---|---|---|---|---|---|
| 0.1 | US | 207 | $4.84_{\pm 0.93}$ | $5.14_{\pm 0.97}$ | $2431.83_{\pm 82.81}$ |
| | CS | 207 | $\mathbf{4.06}_{\pm 0.23}$ | $\mathbf{4.35}_{\pm 0.15}$ | $2430.90_{\pm 70.51}$ |
| 0.2 | US | 214 | $4.16_{\pm 0.77}$ | $4.97_{\pm 0.69}$ | $3306.69_{\pm 5.51}$ |
| | CS | 214 | $\mathbf{3.38}_{\pm 0.31}$ | $\mathbf{3.93}_{\pm 0.47}$ | $3281.46_{\pm 0.86}$ |
| 0.3 | US | 194 | $3.96_{\pm 0.09}$ | $4.84_{\pm 0.18}$ | $3238.40_{\pm 2.68}$ |
| | CS | 194 | $\mathbf{3.74}_{\pm 0.15}$ | $\mathbf{4.65}_{\pm 0.31}$ | $3374.23_{\pm 4.87}$ |

## D.4 Ablation experiments on seeding algorithm

Henceforth, we refer to **seeding initialization** as the approach that uses the $k$-center set produced by Algorithm 1 for initialization. In contrast, **random initialization** refers to selecting $k$ data points uniformly at random from the dataset to form the $k$-center set for initialization.

**Necessity of seeding algorithm on real-world datasets:**

Table 14: Comparison of seeding initialization and random initialization across different numbers of epochs $Z$. We fix the dataset size as 200, $k = 10, \sigma = 1, \tau = 10$ and $\zeta = 0.1$, where $Z$ is the number of iterations in post-processing procedure (i.e., Algorithm 4 in appendix)

| $Z$ | Initialization | cost-nd($\downarrow$) | cost-cd($\downarrow$) | Runtime($\downarrow$) |
|---|---|---|---|---|
| 20 | random | $4.05_{\pm 0.22}$ | $4.62_{\pm 0.97}$ | $280.84_{\pm 0.36}$ |
| | seeding | $\mathbf{2.66}_{\pm 0.05}$ | $\mathbf{3.06}_{\pm 0.37}$ | $322.33_{\pm 0.77}$ |
| 30 | random | $3.95_{\pm 0.65}$ | $4.08_{\pm 0.65}$ | $398.15_{\pm 0.55}$ |
| | seeding | $\mathbf{2.60}_{\pm 0.15}$ | $\mathbf{2.88}_{\pm 0.40}$ | $430.49_{\pm 3.55}$ |
| 50 | random | $3.20_{\pm 0.30}$ | $3.90_{\pm 0.21}$ | $621.08_{\pm 1.34}$ |
| | seeding | $\mathbf{2.33}_{\pm 0.04}$ | $\mathbf{2.55}_{\pm 0.10}$ | $662.62_{\pm 2.18}$ |
| 100 | random | $3.03_{\pm 0.30}$ | $3.17_{\pm 0.29}$ | $1249.73_{\pm 1.69}$ |
| | seeding | $\mathbf{2.35}_{\pm 0.08}$ | $\mathbf{2.56}_{\pm 0.07}$ | $1301.19_{\pm 1.56}$ |
| 150 | random | $2.78_{\pm 0.25}$ | $3.10_{\pm 0.35}$ | $1866.39_{\pm 0.83}$ |
| | seeding | $\mathbf{2.24}_{\pm 0.03}$ | $\mathbf{2.50}_{\pm 0.16}$ | $1900.57_{\pm 0.86}$ |
| 200 | random | $2.30_{\pm 0.03}$ | $2.48_{\pm 0.06}$ | $2381.57_{\pm 64.57}$ |
| | seeding | $\mathbf{2.27}_{\pm 0.10}$ | $\mathbf{2.47}_{\pm 0.13}$ | $2405.65_{\pm 71.16}$ |

Table 14 validates the **effectiveness of our seeding algorithm** (Algorithm 1) for initialization on the ModelNet10 dataset. We randomly select 200 samples from ModelNet10 to form the dataset. In this experiment, initialization is first performed to provide a coarse solution, which is then refined through the post-processing procedure (Algorithm 4) to obtain a finer solution.

To ensure a fair comparison, we account for the computational cost of seeding initialization as equivalent to $k = 10$ epochs. Specifically, in the seeding initialization method, the post-processing algorithm is executed for $Z - k$ iterations, while in the random initialization it runs for $Z$ iterations.

The experimental results demonstrate that our seeding initialization consistently outperforms random initialization in terms of clustering quality. Moreover, it leads to significantly faster convergence, demonstrating the importance of a good initialization strategy.

**Necessity of seeding algorithm on toy dataset:**

We visualized the clean geometric shapes in Figure 11(a), and the noisy geometric shapes are in Figure 3.

Figure 11 validates the **effectiveness of our seeding algorithm** (Algorithm 1) for initialization. In this experiment, we first perform initialization and then apply Algorithm 2 as a post-processing step for clustering. Specifically, Figure 11(b) shows the clustering results with random initialization, while Figure 11(c) presents the results using our Algorithm 1 for initialization. Figure 11(b) shows poor denoising performance without using the seeding algorithm for initialization. In contrast, by using the seeding algorithm, the resulting centers in Figure 11(c) are more closely with the original five clean shapes. This implies the importance of a good initialization, and demonstrates the advantage of our seeding algorithm in providing a good starting point.

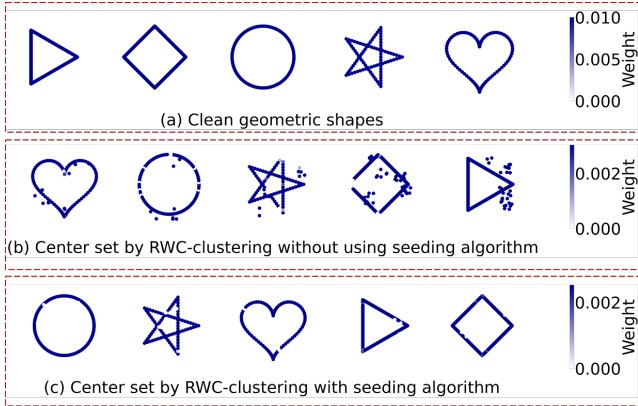

Figure 11: Effectiveness of our seeding algorithm.

