# OpenReview forum: "Robust Wasserstein  $k$-center Clustering: Algorithms and Acceleration"
_NeurIPS.cc/2025/Conference — Submitted to NeurIPS 2025_

### Official Review · Reviewer_x2rS · 2025-06-29

**Clarity:** 3
**Significance:** 2
**Originality:** 2
**Rating:** 4
**Confidence:** 2

**Summary:**

The paper formalises the Robust Wasserstein $k$-center clustering (RWC-clustering) problem, where each data point is a probability measure possibly contaminated by outliers.
The authors argue that comparing a noisy datum $\mu$ to a clean centre $\nu$ using a one-sided robust Wasserstein distance $\mathcal{W}(\mu,\nu)$ preserves intra-cluster similarity, whereas the fully robust distance $\tilde{\mathcal{W}}$ can collapse dissimilar points into the same ball.

---

### Algorithms
A two-stage algorithm is proposed.

The seeding algorithm iteratively picks the point farthest from the current centre set (Gonzalez rule) but purifies each candidate via a “$\tau$-nearest neighbours under $\tilde{\mathcal{W}}$” procedure to strip noise before adding it to the centre set .

Then, a local-search style refinement algorithm repeatedly samples a point proportional to its cost, purifies it, and swaps it with an existing centre when beneficial.

---

### Coreset

In order to improve practical performance, the authors develop some coreset theory for RWC.
Because WW is not a metric, standard coreset theory does not apply. The authors
1. derive a 2-approximate lower bound $\Gamma$ on the optimal radius via a modified Gonzalez routine using $\tilde{\mathcal{W}}$, and
1. build a doubling-metric style coreset whose size is $O\left( \left( \frac{R}{\varepsilon \Gamma} \right)^{2{\rm ddim}} \right)$ and show that any k-centre cost computed on the coreset approximates that on the full set within $\varepsilon$ relative error. Here $R$ is the radius of the dataset.

---

### Experiments
Experiments on small datasets are performed to empirically verify the efficiency and performance of the algorithms.

**Questions:**

1. Do the authors believe an end-to-end algorithm with provable approximation guarantees is possible with their technique?
1. How is the performance of the proposed algorithm on larger instances?

**Ethical Concerns:**

["NO or VERY MINOR ethics concerns only"]

**Final Justification:**

The rebuttal clarified the scalability of the coreset method, which is slightly better than my original understanding.
I believe this could be interesting to the NeurIPS community, but I am not completely convinced of the practicality.
Thus, I maintain my score.

**Limitations:**

yes

**Quality:**

3

**Strengths And Weaknesses:**

### Strengths

1. The main conceptual contribution seems to be a principled formulation for robust clustering in Wasserstein distance, which is an interesting and useful concept.
1. The authors develop a novel coreset algorithm with provable guarantees despite the fact that the objective does not come from a metric, which may be of independent interest
1. Empirical studies include ablation studies of seeding and coreset effectiveness

---

### Weaknesses

1. There is no end-to-end theoretical guarantee of the proposed algorithms. This is not necessarily a weakness if there are extensive experiments. However,
1. the empirical studies seem to be completed on small datasets (size 200), which may not be a good measure of the efficiency of the coreset technique for larger datasets

---

> ### Author Rebuttal · Authors · 2025-07-31
>
> >**1. Do the authors believe an end-to-end algorithm with provable approximation guarantees is possible with their technique?**
>
>
>
> Thank you for your constructive comments.
>
> Providing theoretical guarantees for our Algorithms 1 and 4 is currently quite challenging. However, if we focus solely on theoretical value without considering practical applications, there do exist algorithms with approximation guarantees, although their complexity may be exponential in terms of $k$.
>
> Similar to many constrained clustering algorithms that primarily hold theoretical significance, it is difficult to obtain efficient heuristic algorithms with strong theoretical guarantees in practice. Since our problem additionally involves handling outliers, it is unsurprising that designing efficient heuristics with solid theoretical guarantees is even more challenging.
>
> In future work, we will continue to explore efficient algorithms with theoretical guarantees.
>
> ---
> ---
> ---
>
> >**2. The empirical studies seem to be completed on small datasets (size 200), which may not be a good measure of the efficiency of the coreset technique for larger datasets. How is the performance of the proposed algorithm on larger instances?**
>
>
> Thank you for your insightful comment.
>
> Actually, the dataset size used in the comparative experiments for RWC-clustering with UOT-based clustering and WD-based clustering was 200. However, in our paper, **the coreset method uses a dataset size of 2,500.** We will clarify this clearly in the final manuscript.
>
> To further demonstrate the effectiveness of our approach, we have conducted additional experiments on datasets of varying sizes.
>
> **Table 17**: Comparison of the US method and our CS method using varying dataset size.
> We fix $k = 10$, $Z = 200$, $\sigma = 1$, $\zeta = 0.1$, and $\tau = 10$, where $Z$ is the number of iterations in post-processing procedure (i.e., Algorithm 4 in appendix). The baseline refers to the results obtained on the original dataset.
>
> | Dataset size | Sample method | Sample size | cost-nd  over baseline $\downarrow$        | cost-cd over baseline $\downarrow$        | Runtime  over baseline$\downarrow$         |
> |--------------|----------------|--------------|-------------------------------|-------------------------------|--------------------------------|
> | **1000**     | CS             | 296          | $1.07 _{\pm 0.11}$             | $1.02  _{\pm 0.06}$             | $0.38  _{\pm 0.0006}$            |
> |              | US             | 296          | $1.42  _{\pm 0.06}$             | $1.29  _{\pm 0.08}$             | $0.33  _{\pm 0.0010}$            |
> |--------------|----------------|--------------|-------------------------------|-------------------------------|--------------------------------|
> | **2000**     | CS             | 328          | $1.01  _{\pm 0.07}$             | $0.96  _{\pm 0.05}$             | $0.19  _{\pm 0.0005}$            |
> |              | US             | 328          | $1.25  _{\pm 0.16}$             | $1.33  _{\pm 0.18}$             | $0.15  _{\pm 0.0004}$            |
> |--------------|----------------|--------------|-------------------------------|-------------------------------|--------------------------------|
> | **3000**     | CS             | 314          | $1.22  _{\pm 0.02}$             | $1.23  _{\pm 0.01}$             | $0.15  _{\pm 0.0003}$            |
> |              | US             | 314          | $1.42  _{\pm 0.06}$             | $1.46  _{\pm 0.05}$             | $0.12  _{\pm 0.0001}$            |
> |--------------|----------------|--------------|-------------------------------|-------------------------------|--------------------------------|
> | **4000**     | CS             | 300          | $1.13  _{\pm 0.10}$             | $1.12  _{\pm 0.10}$             | $0.12  _{\pm 0.0036}$            |
> |              | US             | 300          | $1.26  _{\pm 0.14}$             | $1.24  _{\pm 0.12}$             | $0.10  _{\pm 0.0012}$            |
> |--------------|----------------|--------------|-------------------------------|-------------------------------|--------------------------------|
> | **5000**     | CS             | 322          | $1.12  _{\pm 0.09}$             | $1.08  _{\pm 0.07}$             | $0.12  _{\pm 0.0031}$            |
> |              | US             | 322          | $1.33  _{\pm 0.17}$             | $1.33  _{\pm 0.22}$             | $0.09  _{\pm 0.0025}$            |
>
> It can be observed that **across datasets of varying sizes** ranging from 1,000 to 5,000, **our coreset (CS) method consistently outperforms the uniform sampling (US) method.**

---

> > ### Comment · Reviewer_x2rS · 2025-08-02
> >
> > Thanks for the clarification. I believe including this discussion could improve the manuscript. I will maintain my score.

---

> ### Author Response · Authors · 2025-08-02
>
> Thank you for your detailed review and constructive comments.

---

### Official Review · Reviewer_LHf4 · 2025-06-30

**Clarity:** 2
**Significance:** 2
**Originality:** 2
**Rating:** 4
**Confidence:** 5

**Summary:**

In this paper, the RWC-clustering method is proposed to address the shortcomings of the traditional metric k - center clustering in noisy and complex structured data. The approach employs the Robust Wasserstein Distance (RWD) to measure the difference between data points and cluster centers, and designs a dedicated "purification step" to eliminate noise from candidate centers, ensuring the purity of cluster centers. To tackle the computational and storage bottlenecks in large-scale data scenarios, the authors adopt the coreset technique to compress the dataset, theoretically proving that the compressed data can effectively approximate the objective values of the original data, thus enhancing algorithm efficiency. The experiments verify the effectiveness of the RWC clustering algorithm proposed in the paper and the efficiency of the coreset method.

**Questions:**

1.In the steps of the RWC algorithm, both Wasserstein distance and robust Wasserstein distance (RWD) are mixture used. Could the authors elaborate on the specific roles of these two distances in the RWC algorithm respectively?
2.Does the robust Wasserstein distance improve noise immunity by discarding ζ mass, potentially leading to the loss of critical information (e.g., small probability but important features)?
3.Adopting only a single evaluation index and lacking a systematic evaluation of the core indicators of clustering accuracy (e.g., ACC, NMI, ARI, etc.), it is difficult to comprehensively verify the clustering effect of the algorithm.
4.The experimental comparisons are limited to UOT-based and WD-based clustering algorithms, without covering mainstream other baseline methods, making it insufficient to demonstrate the superiority of the proposed algorithm.
5.Only the parameters τ and ζ are ablated individually (as in Tables 11 and 12), without exploring the combined effects of the two on algorithm performance. This lack of systematic exploration of the parameter space may lead to deviations in the selection of optimal parameter combinations.

**Ethical Concerns:**

["NO or VERY MINOR ethics concerns only"]

**Final Justification:**

According to the rebuttal, I will keep my score.

**Limitations:**

The authors acknowledge that the robust Wasserstein distance used does not satisfy the triangle inequality of metric spaces, leading to the inapplicability of theoretical guarantees for traditional clustering algorithms. Although the paper enhances practicality through purification steps and coreset techniques, it lacks a rigorous approximation ratio proof for RWC clustering in non-metric spaces, which renders the theoretical framework incomplete.
Additionally, the construction of coreset relies on the "low doubling dimension" assumption of data, which may not be satisfied by actual complex datasets. The paper fails to fully validate the universality of this assumption in real-world scenarios, nor does it propose coping strategies for performance degradation of coreset in high-dimensional cases—for instance, it does not elaborate on how to adjust parameters to ensure approximation accuracy in high-dimensional contexts.

**Quality:**

2

**Strengths And Weaknesses:**

Strengths：
1.This paper is theoretically sound. The authors analyze the anti-noise mechanism of RWD and the approximation guarantee of coreset compression from the theoretical level, which provides theoretical support for the rationality of the method.
2.This approach demonstrates certain innovation. To address the limitations of traditional k-center clustering in handling noisy and complex-structured data, the RWC-clustering method is proposed, which integrates the Robust Wasserstein Distance (RWD) and a "purification step" to enhance clustering robustness from both the metric design and noise filtering mechanisms.
Weaknesses：
1.The readability of the article is poor. The author provides a series of theoretical analysis, but lacks specific examples to explain the algorithmic process in detail, which would help readers understand the flow of the algorithm.
2.The main text presents only a small number of experimental results, and a large amount of key data (e.g., more dataset comparisons, ablation experiments, etc.) are placed in the appendices, which leads to an incomplete logic of argumentation in the main text, and readers need to rely on the appendices to comprehensively assess the performance of the method, which affects the readability and rigor of the paper.

---

> ### Author Rebuttal · Authors · 2025-07-28
>
> > **1. More illustrations about Wasserstein-related distances**
>
> Thank you for your comments, which have helped improve the readability of our paper.
>
> This paper involves three types of Wasserstein-related distances. For clarity and readability, we assume that outlier mass is $\zeta$ whenever outliers are present. Our method can be easily extended to settings where outliers have different masses.
>
> Given two probability measures $\mu$ and $\nu$:
>
> 1. **Wasserstein distance $W(\mu, \nu)$**: This is the standard Wasserstein distance that directly measures the similarity between $\mu$ and $\nu$ without considering any outliers.
>
> 2. **One-sided robust Wasserstein distance $\mathcal{W}(\mu, \nu)$**: This distance measures the similarity between $\mu$ and $\nu$ after removing outliers of mass $\zeta$ from the first measure $\mu$ only. Importantly, the first argument $\mu$ is denoised, while the second argument $\nu$ remains unchanged.
>
> 3. **Two-sided robust Wasserstein distance $\widetilde{\mathcal{W}}(\mu, \nu)$**: Similar to the one-sided version, but here both measures $\mu$ and $\nu$ are denoised by removing outliers of mass $\zeta$ before computing the distance.
>
> ---
> ---
> ---
>
> > **2. Does the RWD improve noise immunity by discarding ζ mass, potentially leading to the loss of critical information?**
>
> Thank you for your insightful comments.
>
> In fact, as shown in [1,2], the essence of the **robust Wasserstein distance (RWD)** is to **exclude outliers whose total mass does not exceed $\zeta$**. This means that RWD **only removes those points that are explicitly identified as outliers**, and does **not forcibly discard a fixed $\zeta$-mass of points**. This property is also reflected in the experimental results in Table A of our second rebuttal, which directly supports this behavior. It helps to alleviate the concern you raised.
>
> Consequently, our approach **does not require precise prior knowledge of the total outlier mass**. Even if the exact proportion of outliers is unknown, setting $\zeta$ slightly larger does not significantly affect performance. This makes our method **practically robust and easy to apply in real-world scenarios**.
>
>
> ---
>
> [1] Wang X, Huang J, Yang Q, et al. On Robust Wasserstein Barycenter: The Model and Algorithm
>
> [2] Nietert S, Goldfeld Z, Cummings R. Outlier-robust optimal transport: Duality, structure, and statistical analysis
>
> ---
> ---
> ---
>
>
>
> > **3. Other indicators of clustering accuracy (e.g., ACC, NMI, ARI, etc.).**
>
>
>
> Thank you very much for your suggestions.
>
> Indeed, indicators such as ACC, NMI, and ARI are widely used and effective in many scenarios. However, in the context of our research problem, these indicators may not accurately reflect the quality of the clustering results, for the following reasons:
>
> First, our problem is essentially an **unsupervised learning task**, and the datasets we work with **do not necessarily have ground-truth labels**. Even when labels are available, they are often **human-defined** and inherently **subjective**. For the same dataset, different task objectives or interpretations may lead to **different labeling schemes**. These varying schemes can result in significantly different label assignments. However, our clustering algorithms do not utilize label information at all; instead, they group data points based solely on the **intrinsic structure** of the data.
>
> Therefore, there may not be a clear correspondence between our clustering results and the human-provided labels. To illustrate this, consider a simple example with eight 2D points:
>
> $$
> (0,1), (0,-1), (0,2), (0,-2), (0,30), (0,-30), (0,31), (0,-31)
> $$
>
> If we assign labels based on the **sign of the second coordinate**, the following sets emerge:
>
> * "Positive class": { (0,1), (0,2), (0,30), (0,31) }
> * "Negative class": { (0,-1), (0,-2), (0,-30), (0,-31) }
>
> However, if we cluster these points into two groups by solving the **k-center problem in Euclidean space**, a reasonable clustering result might be:
>
> * Cluster 1: { (0,30), (0,-30), (0,31), (0,-31) }
> * Cluster 2: { (0,1), (0,-1), (0,2), (0,-2) }
>
> This clustering result is clearly **not aligned** with the label-based grouping. In this case, using metrics like ACC, NMI, or ARI, which measure the correspondence between cluster assignments and labels, fails to meaningfully reflect the **quality or reasonableness** of the clustering.
>
> Although our clustering problem does not involve labels, it can still **reveal the intrinsic structure of the dataset** very effectively.
>
> ---
> ---
> ---
>
> > **4. Other mainstream baseline methods**
>
> Thank you for your constructive suggestions.
>
> To the best of our knowledge, there has been no prior work directly addressing **robust Wasserstein $k$-center clustering**. Even the baseline methods adopted in this paper—clustering algorithms based on **Unbalanced Optimal Transport (UOT)** and **Wasserstein Distance (WD)**—are **adapted by ourselves**.
>
> Given the limited amount of related research in this direction, we were unable to identify more representative methods. Therefore, the experimental section primarily uses our own adapted approaches as baseline methods.
>
> ---
> ---
> ---
>
>
>
>
>
>
>
> > **5. Explore the combined effects of parameters $\tau$ and $\zeta$.**
>
> Thank you for the constructive suggestions. We further investigated the combined effects of the parameters $\tau$ and $\zeta$.
>
>
> | $\zeta$ | $\tau$ | cost-nd $\downarrow$      | cost-cd $\downarrow$      | Runtime $\downarrow$         |
> |-----------|----------|-----------------------------|------------------------------|--------------------------------|
> | 0.1       | 1        | $10.25 \pm 1.76$           | $10.17 \pm 0.31$            | $2001.51 \pm 7.28$            |
> | 0.1          | 5        | $9.99 \pm 0.55$            | $10.24 \pm 0.40$            | $2104.50 \pm 2.86$            |
> | 0.1          | 10       | $\mathbf{9.86} \pm 1.27$            | $\mathbf{10.19} \pm 0.38$            | $2381.38 \pm 13.68$           |
> | 0.1          | 30       | $11.66 \pm 2.05$            | $10.40 \pm 0.36$            | $4953.87 \pm 6.29$            |
> | 0.2       | 1        | $6.36 \pm 0.83$            | $5.35 \pm 0.52$             | $2497.90 \pm 9.84$            |
> | 0.2          | 5        | $5.51 \pm 1.34$            | $5.25 \pm 0.04$             | $2495.53 \pm 34.97$           |
> |  0.2         | 10       | $\mathbf{5.27 \pm 0.27}$   | $\mathbf{5.18 \pm 0.15}$    | $2742.40 \pm 2.57$            |
> |   0.2        | 30       | $5.83 \pm 0.28$            | $5.01 \pm 0.11$             | $5718.21 \pm 119.58$          |
> | 0.3       | 1        | $3.35 \pm 0.22$            | $2.58 \pm 0.07$             | $2631.57 \pm 23.74$           |
> | 0.3           | 5        | $\mathbf{2.89 \pm 0.35}$   | $\mathbf{2.26 \pm 0.05}$    | $2666.06 \pm 16.76$           |
> |   0.3         | 10       | $2.75 \pm 0.33$            | $2.22 \pm 0.05$             | $2896.39 \pm 34.72$           |
> |    0.3        | 20       | $3.00 \pm 0.73$            | $2.29 \pm 0.11$             | $3799.92 \pm 17.36$           |
> |   0.3         | 30       | $3.15 \pm 0.16$            | $2.28 \pm 0.09$             | $5306.35 \pm 39.59$           |
> | 0.4       | 1        | $3.64 \pm 0.48$            | $2.01 \pm 0.21$             | $2676.09 \pm 27.71$           |
> |  0.4          | 5        | $\mathbf{3.38} \pm 0.74$            | $\mathbf{1.83} \pm 0.08$             | $2776.62 \pm 9.29$            |
> |   0.4         | 10       | $3.39 \pm 0.07$            | $1.79 \pm 0.10$             | $4803.37 \pm 38.40$           |
> |  0.4          | 20       | $3.47 \pm 0.12$            | $1.71 \pm 0.03$             | $3717.48 \pm 23.77$           |
> |   0.4         | 30       | $3.13 \pm 0.19$            | $1.68 \pm 0.05$             | $4571.74 \pm 50.13$           |
> | 0.5       | 1        | $4.15 \pm 0.64$            | $1.73 \pm 0.16$             | $2528.63 \pm 22.15$           |
> | 0.5          | 5        | $3.88 \pm 0.01$            | $1.43 \pm 0.04$             | $2496.95 \pm 1.58$            |
> |  0.5         | 10       | $\mathbf{3.68 \pm 0.18}$   | $\mathbf{1.40 \pm 0.04}$    | $2651.05 \pm 4.15$            |
> |  0.5         | 20       | $4.27 \pm 0.86$            | $1.38 \pm 0.04$             | $3127.46 \pm 2.00$            |
> |  0.5         | 30       | $5.32 \pm 0.17$            | $1.40 \pm 0.03$             | $4223.69 \pm 500.43$          |
> ---
>
> Experimental results show that under different values of $\tau$, when $\zeta$ is smaller than the actual outlier mass (e.g., 0.2,0.1), the performance consistently deteriorates. In contrast, when $\zeta$ is set to 0.3 or slightly higher, its impact on the results is relatively minor.
>
> In addition, setting $\tau$ to 1 leads to suboptimal performance. When $\tau$ is set too high, the performance degrades and computational cost increases significantly. Considering both effectiveness and efficiency, we recommend setting $\tau$ within the range of 2 to 10 in this experiment.
>
> ---
> ---
> ---
>
>
>
>
> > **6. Low doubling dimension assumption of data.**
>
>
> Thank you for your insightful comments. The issue you raised is **crucial to the practical applicability** of our proposed method in real-world scenarios.
>
> In our work, the assumption on the **doubling dimension** is made for the sake of **theoretical guarantees**. In practice, we do **not** require prior knowledge of the exact doubling dimension. Instead, we can simply start with a **relatively small value** and proceed accordingly.
>
> In our experiments, we do not have access to the true doubling dimension of the datasets, which might actually be quite high. Nevertheless, we **uniformly set the doubling dimension to 1**, and the algorithm still performs very well.
>
> This highlights a common **gap between theory and practice**: even when the intrinsic dimension is high, starting with a small value may still lead to effective results in practice. Therefore, our method is **practical and applicable** in real-world scenarios.

---

### Official Review · Reviewer_HivN · 2025-07-02

**Clarity:** 4
**Significance:** 3
**Originality:** 2
**Rating:** 4
**Confidence:** 4

**Summary:**

Robust Wasserstein Center (RWC) clustering is proposed in this paper to address the issue of noise contamination in empirical datasets which standard Wasserstein distances are sensitive to. By adopting a specific variant of robust Wasserstein distance, the RWC-clustering problem is formulated and the algorithm built on the backbone of Gonzalez’s algorithm to select robust Wasserstein centers are proposed.   On the side, a technique that aims to find corsets of the original dataset is proposed to improve the computation efficiency while maintaining accuracy by both theoretically proving that the coreset serves as a set of Wasserstein centers that the RWC-objective is achieved by a negligible distance determined by the number of points in the corset and the optimal value of the RWC-objective.

**Questions:**

It is assumed in the paper that $\mu$ and $\nu$ both have $\zeta$ mass of outliers: is there any specific reason for imposing this assumption and does it change the conclusion if two distributions are allowed to have different mass of outliers? And how is $\zeta$ chosen in reality if it is unkown?

In terms of the computation of robust Wasserstein distance, how is it handled given that it is a minimum of Wasserstein distance over a set of distributions and the computation of Wasserstein distance (especially in high dimensions) are known to be inefficient?

**Ethical Concerns:**

["NO or VERY MINOR ethics concerns only"]

**Final Justification:**

I would keep my original recommendation for the rating, i.e. 4.
The authors have resolved my questions concerning the extension to different noise mass as well as the computation inefficiency of empirical Wasserstein distances.
On the other hand, the empirical experiment/computation around various choices of the noise mass partially addressed my question around the appropriate choice of the noise mass when it is unknown, which seems to be one of the questions from the other reviewers as well. However, the response "slightly larger than the actual noise mass" is not only unrigorous, but also impractical in applications.
Considering the answers to my questions (which have some overlap with discussions between the authors and the other reviewers) and the overall novelty and significance of the paper, I would recommend a borderline accept.

**Limitations:**

There is no major limitations but the concerns mentioned in the questions above may limit the effectiveness and broadness of the applications to real-world problems.

**Quality:**

4

**Strengths And Weaknesses:**

Strengths:
1. The paper is very comprehensive and clearly written by organizing and sequencing the problem formulation, intuition, theories, algorithms, experiments, and vivid simple examples.
2. The proof is rigorous and solid.
3. The experiments are very detailed and well presented, serving as strong arguments for the proposed algorithms.
4. The authors are very thorough in considering the whole pipeline from both theoretical and computational perspectives and have made valid points in each involved steps.

Weakness:
1. While there is innovation and good power of broad applications in the proposal of RWC clustering and the associated algorithms, a big portions of the underlying theories and the algorithms seem to stem from and be built on previous related work without much novel design and/or theoretical findings.

---

> ### Author Rebuttal · Authors · 2025-07-27
>
> > **1. It is assumed in the paper that $\mu$ and $\nu$ both have $\zeta$ mass of outliers: is there any specific reason for imposing this assumption and does it change the conclusion if two distributions are allowed to have different mass of outliers?**
>
> Thank you for your insightful comments, which have helped us further improve the paper and enhance its readability.
>
> In the paper, we assume that both \$\mu\$ and \$\nu\$ share the same outlier mass \$\zeta\$, primarily for the sake of clarity and to improve the readability of the presentation.
>
> As indicated in \[1,2], **our method and theoretical analysis can be naturally extended to the more general case where \$\mu\$ and \$\nu\$ have different outlier masses, without affecting the core conclusions.** We will make this point explicit in the final version to avoid any potential misunderstandings.
>
> ---
>
> [1] Wang X, Huang J, Yang Q, et al. On Robust Wasserstein Barycenter: The Model and Algorithm[C]//Proceedings of the 2024 SIAM International Conference on Data Mining (SDM). Society for Industrial and Applied Mathematics, 2024: 235-243.
>
> [2] Nietert S, Goldfeld Z, Cummings R. Outlier-robust optimal transport: Duality, structure, and statistical analysis[C]//International Conference on Artificial Intelligence and Statistics. PMLR, 2022: 11691-11719.
>
> ---
> ---
> ---
>
>
> > **2. And how is $\zeta$ chosen in reality if it is unkown?**
>
> We sincerely appreciate your insightful comment, which nicely points out one of the advantages of our method.
>
> As shown in [2], when the parameter \$\zeta\$ is set slightly larger than the actual noise mass, the optimal value can still be attained in theory. This suggests that a mild overestimation of \$\zeta\$ does not significantly affect the results.
> To validate this observation, we conducted the following experiments.
>
>
> We randomly sample 200 samples from the ModelNet10 to form the dataset, each data item containing 0.3 mass of noise. However, we run our algorithm across different \$\zeta\$.
>
> **Table A**: Comparison of our algorithm (i.e., Algorithm 1 and 4) for solving **RWC-clustering** problem across different values of \$\zeta\$ on ModelNet10 dataset with 0.3 mass of noise. We fix the dataset size as 200, \$k = 10\$, \$Z = 200\$, \$\sigma = 1\$ and \$\tau = 10\$, where \$Z\$ is the number of iterations in post-processing procedure (i.e., Algorithm 4).
>
>
> ---
> | \$\zeta\$ |       cost-nd ($\downarrow$)      |       cost-cd ($\downarrow$)      |       Runtime ($\downarrow$)       |
> | :-------: | :--------------------: | :--------------------: | :---------------------: |
> |    0.05   |   13.48\$_{\pm0.15}\$  |   13.58\$_{\pm1.48}\$  |  2387.10\$_{\pm8.16}\$  |
> |    0.1    |   9.99\$_{\pm0.11}\$   |   10.90\$_{\pm1.51}\$  | 2861.19\$_{\pm189.57}\$ |
> |    0.2    |   5.10\$_{\pm0.02}\$   |   5.45\$_{\pm0.69}\$   | 2985.63\$_{\pm226.70}\$ |
> |    **0.3**    | **2.29\$_{\pm0.17}\$** | **3.58\$_{\pm0.94}\$** | 2998.34\$_{\pm166.96}\$ |
> |    0.4    |   1.72\$_{\pm0.07}\$   |   3.26\$_{\pm0.36}\$   |  2943.35\$_{\pm35.42}\$ |
> |    0.5    |   1.43\$_{\pm0.08}\$   |   3.86\$_{\pm0.49}\$   |  3055.85\$_{\pm88.55}\$ |
> |    0.6    |   1.21\$_{\pm0.06}\$   |   5.01\$_{\pm0.59}\$   | 3122.77\$_{\pm112.56}\$ |
> |    0.7    |   1.00\$_{\pm0.04}\$   |   4.46\$_{\pm0.37}\$   |  3589.76\$_{\pm26.52}\$ |
> |    0.8    |   0.75\$_{\pm0.01}\$   |   4.93\$_{\pm0.08}\$   |  3722.17\$_{\pm16.04}\$ |
> |    0.9    |   0.59\$_{\pm0.02}\$   |   7.30\$_{\pm1.00}\$   |  3921.55\$_{\pm14.58}\$ |
> ---
>
>
>
> The results confirm that **a slight overestimation of \$\zeta\$ only leads to a minor impact on performance**, whereas underestimating it can severely degrade the solution quality.
>
> Therefore, **when the actual noise mass is unknown, we recommend setting \$\zeta\$ slightly larger than the expected noise level**.
>
> This feature allows our method to work well without a precise specification of the outlier mass, making it and particularly practical in real-world scenarios.
>
>
>
> ---
> ---
> ---
>
>
>
> > **3. In terms of the computation of robust Wasserstein distance, how is it handled given that it is a minimum of Wasserstein distance over a set of distributions and the computation of Wasserstein distance (especially in high dimensions) are known to be inefficient?**
>
> We sincerely appreciate your insightful comment, which exactly points out another advantage of our method.
>
> Based on the result in [1], the robust Wasserstein distance (RWD) can be reduced to an augmented optimal transport (OT) problem of the same scale. In other words, for two probability measures \$\mu\$ and \$\nu\$ with support sizes of \$O(n)\$, the robust Wasserstein distance between them can be computed by solving an OT problem whose support size is also \$O(n)\$.
>
> This is particularly important because it means we can directly leverage existing OT solvers to compute the RWD. The OT problem has been extensively studied, and many efficient solvers are available. **More importantly, this also implies that most of the existing OT acceleration techniques can be directly applied to our method, and future algorithmic advances in the OT field will naturally enhance the performance of our approach.**

---

> > ### Comment · Reviewer_HivN · 2025-08-06
> >
> > Thank you for the detailed clarification especially the numbers in q2! Considering the overall novelty and significance, I will maintain my score.

---

### Official Review · Reviewer_RXqT · 2025-07-05

**Clarity:** 1
**Significance:** 1
**Originality:** 2
**Rating:** 2
**Confidence:** 4

**Summary:**

This paper studies the k-center problem under the Wasserstein distance. The Wasserstein distance (also known as the Earth Mover's Distance) measures the cost of optimally transporting mass to transform one distribution into another. While it is widely used in machine learning and computational geometry, it is important to note that the Wasserstein distance is generally not a metric over the space of discrete distributions when the supports differ. This raises questions about how traditional metric-space clustering algorithms, such as k-center, can meaningfully be applied.

I found the paper difficult to follow. I have worked on the k-center problem for almost 20 years, but the technical exposition here is quite difficult to understand. From what I understand, the authors propose adding a purification step before running the Gonzalez algorithm. Specifically, Algorithm 1 on page 5 simulates the Gonzalez procedure, and in each iteration it selects the tau closest points to the most recently chosen center to find a better candidate center among them. However, on page 6 the authors write “we choose tau that retain the smallest tau for Equation 5.” I do not understand what this means, why it is justified, or how it fits into the algorithm.

Moreover, the paper does not mention the approximation factor achieved by the proposed algorithm in the “Our Contribution” section on page 2, which is a crucial omission. In fact, this section is generally unclear. For example, the authors write “For effectively representing datasets with complex structures and outliers, we introduce the RWC-clustering problem and provide the underlying intuition for its formulation.” However, the terms “complex structures” and “outliers” are not clearly defined, making it hard to understand what precise problem the authors are solving.

Later, the paper states: “To solve this robust clustering problem, we first design a purification step to eliminate the noise in the candidate centers; then, we integrate it into existing methods [Gonzalez 1985; Lattanzi and Sohler 2019; Choo et al. 2020] to develop customized initialization and post-processing algorithms.” However, the purification step itself is not clearly explained or sketched in intuitive terms in the “Our Contribution” section. Furthermore, it seems the authors may not be aware that the result of Lattanzi and Sohler 2019 is for k-means clustering, not k-center clustering.

The paper also mentions: “Furthermore, to enhance scalability, we introduce the coreset technique to accelerate the computation by compressing the dataset. Additionally, we theoretically demonstrate that the coreset is a good proxy of the original dataset.” However, the construction of the coreset is not clearly described in the “Our Contribution” section, and the theoretical guarantees are not stated precisely. Importantly, the coreset construction appears to rely on bounded doubling dimension, which is a property defined for metric spaces. Since the Wasserstein distance here is generally not a metric, this assumption should be explicitly acknowledged and justified.

In summary, the paper needs significant polishing and clarification:
- First, clearly define what is meant by “complex structures and outliers.”
- Next, explicitly state the main theorems, approximation factors, and coreset guarantees achieved by the algorithm.
- Finally, rewrite the “Our Contribution” section to clearly and concretely describe the proposed method and why it is novel.

Addressing these issues would greatly help readers understand the motivation and technical content of the paper.

**Questions:**

I have mentioned them in above.

**Ethical Concerns:**

["NO or VERY MINOR ethics concerns only"]

**Limitations:**

Mentioned in above.

**Paper Formatting Concerns:**

References are not in common format.

**Quality:**

1

**Strengths And Weaknesses:**

**Weaknesses**
- The paper is difficult to follow: key ideas (e.g., purification step, coreset construction) are not clearly explained.
- Lacks precise statements of approximation factors, theoretical guarantees, and assumptions.
- The related work discussion is incomplete and occasionally inaccurate (e.g., citing k-means results for k-center).
- Relies on bounded doubling dimension without explaining how this applies under the Wasserstein distance, which is generally not a metric.

---

> ### Author Rebuttal · Authors · 2025-07-25
>
> > **1. Clarification of background**
>
> We sincerely appreciate your insightful feedback. We apologize for the previous ambiguity and would like to clarify our motivation more precisely.
>
> Our work focuses on the **$k$-center clustering problem under the Robust Wasserstein Distance (RWD)**, referred to as the **Robust Wasserstein $k$-Center Clustering ($k$-RWC-clustering)** problem.
>
> In the classical setting, the **Wasserstein distance is a proper metric** on the space of probability measures, which enables many existing $k$-center algorithms and theoretical results—developed under the metric space assumption—to be directly applied.
>
> However, in its **robust scenarios**, the **Robust Wasserstein Distance (RWD) no longer satisfies metric properties**, such as the **triangle inequality**.
>
> As a result, traditional theoretical analyses based on metric assumptions **can no longer be directly applied** in our robust version, giving rise to **substantial algorithmic and theoretical challenges**.
>
> ---
> ---
> ---
>
> > **2. Poblems on doubling dimension**
>
> Thank you for your valuable suggestions on improving the readability of our paper.
>
> In our work, the assumption of low doubling dimension is introduced under the **Wasserstein distance**, which **is a metric** on the space of probability measures. This makes our assumption well-founded.
>
> Moreover, for any two probability measures, their robust Wasserstein distance (RWD) is always no greater than their Wasserstein distance. As a result, a coreset constructed under the Wasserstein distance naturally forms an $r$-cover of the original dataset in terms of the RWD, effectively capturing and representing the original dataset in our robust scenarios.
>
> **We use the Wasserstein distance for coreset construction primarily because it enables theoretical guarantees.** In practice, **data often exhibit intrinsic structures, making the low doubling dimension assumption reasonable and commonly satisfied in many real-world scenarios.**
>
> More importantly, **our algorithm does not require prior knowledge of the doubling dimension.** It suffices to start from a small value and proceed adaptively. In our experiments, setting the doubling dimension to 1 already yields satisfactory coreset performance. Therefore, even when the doubling dimension is unknown, the algorithm remains effective in practice, which **makes our acceleration method broadly applicable in real-world scenarios.**
>
> ---
> ---
> ---
>
> > **3. Complex structures and outliers**
>
> **Complex structures**
>
> Many datasets can be conveniently represented as vectors, matrices, or tensors, and their similarities can be measured using Euclidean distance. However, for certain types of data—such as **point clouds, graphs, or other highly structured objects**—Euclidean distance fails to effectively capture the differences between instances. In this work, we refer to such data as **complex structured data**. These data types often exhibit significant structural and geometric properties that traditional distance metrics may overlook. In such cases, Wasserstein-based distances are more suitable for measuring similarity.
>
>
> **Outliers**
>
> In real-world scenarios, data collection is often subject to noise caused by various factors such as sensor errors, environmental fluctuations, or human operational mistakes. These errors can introduce **outliers**—data points that deviate significantly from the main distribution. Such outliers may heavily distort the analysis or learning results if not properly handled. Therefore, it is crucial to explicitly consider the presence of outliers when designing algorithms for real-world applications, in order to ensure the stability and robustness of the model.
>
> ---
> ---
> ---
>
> > **4. Why Lattanzi and Sohler (2019)**
>
> Thank you for your insightful guidance on our approach.
>
> Lattanzi and Sohler (2019) proposed an efficient local search algorithm for the $k$-means problem. Although originally developed for $k$-means, its simplicity, computational efficiency, and independence from properties such as the triangle inequality make it particularly suitable for our robust clustering scenarios. We adopt their local search framework and adapt it to our robust clustering setting. Combined with our customized purification step, we employ it as a post-processing algorithm (Algorithm 4) to further refine clustering result.
>
> ---
> ---
> ---
>
> > **5. More illustration** on "..., we retain the smallest \(\tau\) values of \(\widetilde{\mathcal{W}}(\cdot, \nu)\) in \Cref{Eq:pure-2}, rather than selecting only one." on page 5.
>
> I'm very sorry for causing the misunderstanding.
>
> While fixing $\tau = 1$ would still allow the algorithm to work, we choose to make $\tau$ a tunable parameter and retain the $\tau$ smallest values of $\widetilde{\mathcal{W}}(\cdot, \nu)$ in Equation (5). This enhances the algorithm’s flexibility and often leads to better empirical performance.
>
> ---
> ---
> ---
>
>
> > **6. Our contribution**
>
> - We introduce the **RWC-clustering** problem to handle datasets with complex structures and noise.
> - We design heuristic algorithms to solve the RWC-clustering problem, including an **initialization algorithm** (Algorithm 1) and a **post-processing algorithm** (Algorithm 4). Since the **RWD used in our definition is non-metric**, this brings significant challenges to theoretical analysis. Although we currently do not provide approximation guarantees, we validate the effectiveness of our algorithms on multiple datasets.
>
> - To improve scalability, we incorporate the **coreset** technique for data compression. By running the algorithms on a small coreset instead of the full dataset, we can significantly accelerate the computation. This method is supported by theoretical guarantees and shows strong empirical performance.
>
>
>
> ---
>
> **Algorithm for the RWC-clustering problem**
>
> Our proposed algorithm consists of two main stages: an **initialization algorithm** (Algorithm 1) and a **post-processing algorithm** (Algorithm 4). The core idea is to integrate a **purification step** into a existing clustering framework. The entire approach is heuristic, tailored to handle datasets with noisy and structured characteristics.
>
> ---
>
> **Motivation for the purification step**
>
> Our objective is to obtain **clean cluster centers**. In classical metric $k$-center clustering, it is usually assumed that the input dataset is noise-free, and thus, selecting centers directly from the dataset suffices. However, in robust clustering scenarios—such as ours—the dataset may contain outliers, and directly chosen centers may be **contaminated**. To address this, we introduce a **purification step** that removes outliers from the selected candidates, yielding clean cluster centers.
>
> ---
>
> **More details of the purification step**
>
> Given a candidate center $\nu$ selected from the dataset, we perform a purification step to derive its corresponding clean center $\tilde{\nu}$ by removing outliers.
>
> Ideally, we hope to find a data point $\mu$ in the dataset such that, after removing outliers from both $\mu$ and $\nu$, the cleaned versions are **identical**, i.e.,$\widetilde{\mathcal{W}}(\mu, \nu) = 0.$
>
> However, such an ideal match rarely exists in practice, and $\mu$ may not be unique.
>
> Thus, we relax this goal: for each candidate center $\nu$, we compute the top-$\tau$ points $\mu$ in the dataset that minimize $\widetilde{\mathcal{W}}(\mu, \nu)$. This process yields $\tau$ purified versions of $\nu$. Among them, we select the best-performing one as our clean center.
>
>
>  **Scalability via the coreset technique (Algorithm 2)**
>
> To accelerate the clustering process, we incorporate a **coreset-based data compression technique** into our framework. Specifically, we use a **hierarchical Gonzalez algorithm** to construct a compact and representative summary of the original dataset.
>
> This coreset allows us to run the clustering algorithm on a much smaller subset while approximating the results on the full dataset. To ensure theoretical guarantees, we assume that the dataset has bounded **doubling dimension** under the **Wasserstein distance**. The main technical challenge lies in the theoretical analysis（in Appendix C.3).
>
> ---
> ---
> ---
>
> > **7. Main theorems, approximation factors, and coreset guarantees**
>
> Our work consists of two main components: solving the **RWC-clustering** problem (Algorithm 1 and 4), and accelerating the solution using a **coreset-based technique** (Algorithm 2).
>
> **Solving the RWC-clustering problem (Algorithm 1 and 4)**
>
> The core challenge in analyzing the RWC-clustering problem arises from the fact that the **Robust Wasserstein Distance (RWD)** used in our formulation is **not a metric**, which makes standard approximation techniques difficult to apply. As a result, we currently can not offer a theoretically guaranteed approximation for our Algorithm 1 and 2. However, we have extensively validated the effectiveness of our algorithms on multiple datasets.
>
> **Acceleration via coreset technique (Algorithm 2)**
>
> In contrast, our **acceleration strategy using coresets** enjoys rigorous theoretical guarantees. By introducing a novel analysis framework (in Appendix 4.3), we prove our main result in **Theorem 4.2**. Specifically, assuming the data has a **doubling dimension** of $\mathsf{ddim}$, Algorithm 2 constructs an $\epsilon$-coreset of size $O((\frac{R}{r})^{2\cdot \mathsf{ddim}})$.
>
> Roughly speaking, for any $k$-center set in the solution space, the clustering cost evaluated on the original dataset can be well approximated—**within a relative error of $\epsilon$**—by the cost evaluated on the coreset. This implies that the coreset serves as a **compact and faithful representation** of the original dataset with respect to the RWC-clustering objective. Consequently, we can solve the RWC-clustering problem efficiently on the small coreset, while still achieving a good approximation to the solution on the full dataset.

---

### Note · Authors · 2025-08-15

We sincerely thank everyone for the time and effort they have devoted to our paper.

We are not aware of the details of the Reviewer–AC Discussion phase.

However, based on the information available to us, it is evident that **Reviewer RXqT** has a major misunderstanding of our work. This is clear from the first paragraph of his review:

> "This paper studies the **$k$-center problem under the Wasserstein distance**. $\cdots \cdots$ it is important to note that the **Wasserstein distance is generally not a metric** over the space of discrete distributions when the supports differ. This raises questions about how traditional metric-space clustering algorithms, such as $k$-center, can meaningfully be applied."

Actually, the core of our work is **not** the *$k$-center problem under the Wasserstein distance*, but rather its **robust version**. This can be easily seen from both the title and the abstract of our paper. If this misunderstanding exists, the main challenge of our work—**designing algorithms for the robust version**—will be ignored. In such a case, it is difficult for the reviewer to follow the subsequent algorithms and analyses.

Furthermore, the **Wasserstein distance itself is indeed a metric** [1], a point we explicitly mention in line 111 of our paper. The reviewer’s claim that “the Wasserstein distance is not a metric” is incorrect. What actually fails to satisfy the metric properties is the **Robust Wasserstein Distance (RWD)**.

In Section 4, for the sake of obtaining rigorous theoretical guarantees, we assume the data has a finite **doubling dimension under the Wasserstein distance**. If the reviewer mistakenly believes that “the Wasserstein distance is not a metric,” he may wrongly believe that the theoretical derivations in Section 4 are fundamentally flawed.

I strongly suspect that, due to the above misunderstanding, this reviewer assigned us a **score of 2**.

During the discussion phase, I never received a substantive response from this reviewer. I am unsure whether he actually read my rebuttal or whether he had further questions.

It is normal for misunderstandings to occur in reviews. However, at this point, I still have not seen this reviewer’s final rating. Unless he has been dealing with urgent matters throughout this period, I think he is irresponsible.

---
[1]Peyré G, Cuturi M. Computational optimal transport: With applications to data science[J]. Foundations and Trends® in Machine Learning, 2019, 11(5-6): 355-607.

---

### Decision · Program_Chairs · 2025-09-17

**Decision:**

Reject

**Comment:**

This paper proposes a Robust Wasserstein Center clustering method, which heuristically combines a few stages including a purification step followed by a new coreset construction that is meant to be robust under Wasserstein distance. Several referees find the formulation conceptually interesting and the empirical results show some promise. However, reviewers lean borderline with one reviewer remaining more concerned about weaknesses that I share after reading the paper. In short, the paper is not written with the level of clarity and precision compared with high quality submission to NeurIPS. Key components of the algorithm are not well explained or precisely defined, and the overall heuristic idea is a good one in my opinion, but needs to be supported explicitly, with precise theoretical framing. I encourage the authors to prepare a stronger revision given the constructive comments of the referees, though the paper would benefit greatly from a further round of review.